# Functional profiling of single CRISPR/Cas9-edited human long-term hematopoietic stem cells

Elvin Wagenblast [1], Maria Azkanaz[1], Sabrina A. Smith[1], Lorien Shakib[1], Jessica L. McLeod[1], Gabriela Krivdova[1], Joana Araújo[1], Leonard D. Shultz[2], Olga I. Gan[1], John E. Dick [1,3]* & Eric R. Lechman[1]*

In the human hematopoietic system, rare self-renewing multipotent long-term hematopoietic stem cells (LT-HSCs) are responsible for the lifelong production of mature blood cells and are the rational target for clinical regenerative therapies. However, the heterogeneity in the hematopoietic stem cell compartment and variable outcomes of CRISPR/Cas9 editing make functional interrogation of rare LT-HSCs challenging. Here, we report high efficiency LT-HSC editing at single-cell resolution using electroporation of modified synthetic gRNAs and Cas9 protein. Targeted short isoform expression of the GATA1 transcription factor elicit distinct differentiation and proliferation effects in single highly purified LT-HSC when analyzed with functional in vitro differentiation and long-term repopulation xenotransplantation assays. Our method represents a blueprint for systematic genetic analysis of complex tissue hierarchies at single-cell resolution.

[1] Princess Margaret Cancer Centre, University Health Network, Toronto, ON M5G 1L7, Canada. [2] The Jackson Laboratory, Bar Harbor, ME 04609, USA. [3] Department of Molecular Genetics, University of Toronto, Toronto, ON M5S 1A8, Canada. These authors contributed equally: John E. Dick, Eric R. Lechman. *email: john.dick@uhnresearch.ca; elechman@uhnresearch.ca

The hematopoietic stem and progenitor cell (HSPC) compartment is a functional continuum comprised of multiple stem and progenitor cell populations including abundant committed progenitors such as common myeloid progenitors and myelo–erythroid progenitors (MEPs), as well as rarer multipotent stem cells including short-term hematopoietic stem cells (ST-HSCs) and long-term hematopoietic stem cells (LT-HSCs)[1,2] (Supplementary Fig. 1). LT-HSCs are the only population that have the ability to permanently repopulate the entire hematopoietic system following transplantation[2]; they represent the key target for blood-based regenerative therapies. Thus, LT-HSCs are essential for therapeutic genome editing to correct acquired and genetic hematopoietic disorders[3,4]. Furthermore, the pathogenesis of hematological malignancies like acute myeloid leukemia (AML) is associated with the presence of initiating mutations acquired in LT-HSCs, which lead to their competitive expansion[5,6]. Pre-leukaemic LT-HSCs are a source of clonal evolution within blood malignancies and can act as a reservoir of relapse after chemotherapy treatment[7]. Previous studies have shown the feasibility of using genome editing techniques in human CD34+ cells to model hematological diseases, such as using CRISPR/Cas9 to induce myeloid neoplasia[8] or using transcription activator-like effector nucleases to induce chromosomal translocations to model MLL-rearranged leukemia[9]. Thus, modeling and understanding the genetic complexity and cellular heterogeneity seen in human hematological malignancies using novel methodologies that allow genome editing in highly purified single LT-HSCs and their functional read-out are of great need[3,4].

Recently several studies have demonstrated efficient gene editing of bulk CD34+ populations that are enriched for human HSPCs[8,10–18]. Highly efficient non-homologous end joining (NHEJ)-mediated gene disruption of up to 80–90% efficiency has been reported in CD34+ HSPCs[12,13,15,17]. In addition, homology-directed repair (HDR)-mediated knock-ins, with or without selectable fluorescent reporter genes, have been established with an efficiency of up to 20% in CD34+ HSPCs[10,11,13,14,16,18]. Stable integration of a fluorescent reporter using rAAV6 combined with flow cytometry-based sorting enabled enrichment of CRISPR/Cas9-edited HSPCs[10,11,17]. Because LT-HSCs represent only 0.1–1% of CD34+ populations, these studies did not address LT-HSC targeting in the most direct manner. Previous studies have reported long-term engraftment of up to 16 weeks following xenotransplantation of CRISPR/Cas9 edited human CD34+ HSPCs[10,16,18], suggesting that rare LT-HSCs within the CD34+ population can be gene edited. However, these studies utilized considerable numbers of CD34+ HSPCs and lacked the resolution to functionally interrogate the differentiation and proliferation properties of individual LT-HSCs. In order to simplify LT-HSC targeting within heterogeneous CD34+ HSPCs, we explored the possibility of CRISPR/Cas9 editing in highly purified LT-HSCs. This approach would enable direct functional characterization of LT-HSCs, rather than bulk populations. Here, we show successful editing of highly purified LT-HSCs via CRISPR/Cas9-mediated NHEJ or HDR and their subsequent functional investigation using single cell in vitro differentiation and near-clonal xenotransplantation assays.

## Results

### CRISPR/Cas9-mediated GATA1 isoform expression in LT-HSCs.
As a proof of principle, we modeled GATA1 isoform expression in LT-HSCs, ST-HSCs, and MEPs from neonatal cord blood using CRISPR/Cas9. GATA1 encodes a DNA binding protein with two zinc fingers and a transactivation domain that is required for erythroid, megakaryocyte, mast cell, eosinophil, and basophil differentiation[19–21]. Acquired and inherited GATA1

mutations contribute to hematological disorders such as Down syndrome acute megakaryoblastic leukemia (AMKL), Diamond-Blackfan anemia, transient myeloproliferative disorder and congenital dyserythropoietic anemias with thrombocytopenia[22–27]. The GATA1 gene normally produces two protein isoforms as a result of alternative mRNA splicing—the GATA1 full length (GATA1-Long) and a truncated form (GATA1-Short). GATA1 isoform biology is particularly important for children with Down syndrome and a subset of children born with Diamond-Blackfan anemia. Children with Down syndrome have a 150-fold higher risk of developing AMKL, which is characterized by an abnormal proliferation of immature megakaryocytes[28–30]. Mutations in exon 2, which lead to the exclusive expression of GATA1-Short, are thought to be an essential driver of this disease. Previous work has shown effects of GATA1-Short on megakaryocytic proliferation, but these changes were only seen in fetal HSPCs and not in neonatal or adult HSPCs, implying a developmental stage-specific effect[31,32]. To verify this hypothesis, we decided to test GATA1-Long versus GATA1-Short isoform expression in purified LT-HSCs, ST-HSCs, and MEPs from neonatal cord blood.

Overall, our experimental scheme employed flow cytometric isolation of cord blood LT-HSCs for xenotransplantation and isolation of LT-HSCs, ST-HSCs, and MEPs for single cell in vitro differentiation assays (Supplementary Fig. 2). Sorted cells were cultured for 48 h and electroporated with modified synthetic gRNAs and Cas9 protein (Fig. 1a). During brief in vitro culturing, LT-HSCs durably retain their immuno-phenotype with some variability seen in ST-HSCs and MEP subsets. (Supplementary Fig. 3). Because of the transient delivery of the ribonucleoprotein complex and lack of a selectable marker, all CRISPR/Cas9-mediated edits were subsequently genetically verified. To express the GATA1-Short isoform, we used two gRNAs targeting the 5′ and 3′ flanking regions of exon 2, resulting in the NHEJ-mediated dropout of the exon (Fig. 1b). By contrast, mutation of the GATA1-Short alternative start site on exon 3 from ATG to CTC via CRISPR/Cas9-mediated HDR led to the exclusive expression of the GATA1-Long isoform (Fig. 1c). Importantly, expression of both the GATA1-Short and -Long isoforms remained under the regulatory control of the endogenous GATA1 promoter. Because GATA1 is X-linked, all our studies utilized male cord blood samples. As a control, we used two gRNAs targeting exon 1 of the olfactory receptor OR2W5 that were designed with the CRoatan algorithm[33], resulting in a 150 bp dropout of the exon. After electroporation, individual LT-HSCs, ST-HSCs, and MEPs were deposited into single cell in vitro assays under erythro-myeloid differentiation conditions[34] (Fig. 1a). After 16–17 days, each single cell-derived colony was assessed by flow cytometry for lineage output and the genotype of GATA1-Short or control edited cells was determined by polymerase chain reaction (PCR), whereas the genotype of GATA1-Long edited cells was assessed by Sanger sequencing (Supplementary Fig. 4a–c). Moreover, LT-HSCs that were CRISPR/Cas9 edited with GATA1-Short or control gRNAs were transplanted into mice at a near-clonal level in order to detect lineage and proliferation biases (Fig. 1a).

### Single-cell differentiation of CRISPR/Cas9-edited LT-HSCs.
CRISPR/Cas9 editing efficiency in LT-HSCs, ST-HSCs, and MEPs was high; the percentage of single cell-derived colonies with homozygous deletion of OR2W5, GATA1-Short, and GATA1-Long was 50–60%, 40%, and 20%, respectively (Fig. 2a and Supplementary Fig 5a). Any control or GATA1-Short edited single cell colony with only one gRNA cut and no exon dropout was disregarded in the initial analysis. Although statistically insignificant, CRISPR/Cas9-mediated HDR efficiencies were lower in LT-HSCs than in ST-HSCs and MEPs (Fig. 2a). While

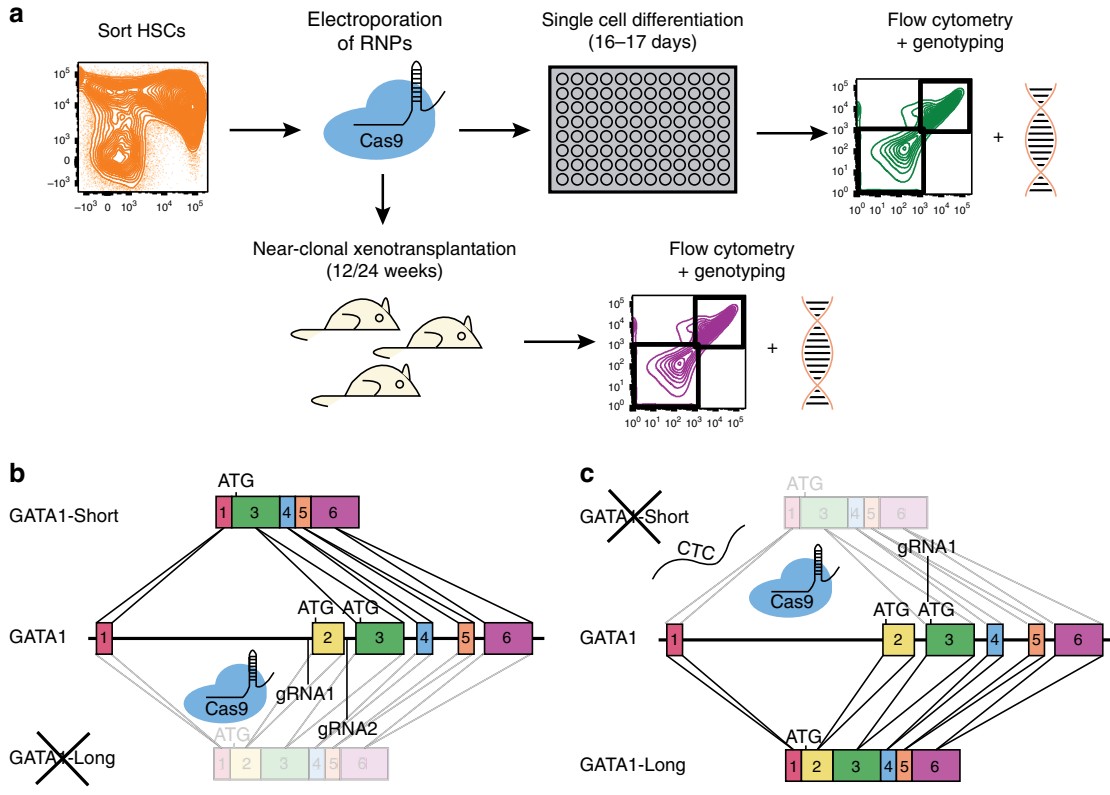

**Fig. 1** CRISPR/Cas9-mediated isoform expression of GATA1 at single cell level. **a** Experimental workflow for single cell in vitro differentiation assay and near-clonal xenotransplantation into mice. **b** Two gRNAs targeting the 5′ and 3′ end of exon 2 of GATA1 led to the NHEJ-mediated dropout of this exon, resulting in the exclusive expression of GATA1-Short. **c** HDR-mediated mutation of the alternative start site from ATG to CTC using a single gRNA and a single-strand DNA template, resulting in the exclusive expression of GATA1-Long

electroporation by itself did not drastically affect the efficiency of single cells to form colonies, LT-HSCs had slightly lower single cell colony formation efficiencies compared to ST-HSCs and MEPs (Fig. 2b). No off-target cleavage was detected at loci that were similar in sequence to the gRNA target sequence by amplicon Sanger sequencing (Supplementary Fig. 5b). Although no whole genome sequencing was performed, the likelihood that these results are due to off-target cleavage is extremely low. In addition, karyotyping analysis revealed no structural abnormalities after CRISPR/Cas9 editing in any of the conditions (Supplementary Fig. 5c). Western assay of bulk CRISPR/Cas9-edited MEPs cultured under erythro-myeloid conditions showed enrichment of either the GATA1-Long or GATA1-Short isoform (Fig. 2c, Supplementary Fig. 6a).

Culture of single LT-HSC, ST-HSC, and MEP under erythro-myeloid differentiation conditions revealed a drastic shift towards megakaryocytic lineage output upon exclusive expression of GATA1-Short, with a twofold and fourfold increase in $CD41^+$ megakaryocytic colonies compared to LT-HSCs expressing control and GATA1-Long, respectively (Fig. 2d, e). Interestingly, only GATA1-Short edited LT-HSCs produced bi-potent myelo-megakaryocytic colonies. ST-HSCs showed even higher fold increases toward megakaryocytic lineage output compared to LT-HSCs. Whereas CRISPR/Cas9 control edited MEPs did not possess any megakaryocytic differentiation capacity, GATA1-Short edited MEPs were able to produce $CD41^+$ megakaryocytes, albeit with lower efficiency compared to LT-HSCs and ST-HSCs. In order to mimic precise mutations at the 5′ splice junction of exon 2 of patients with Down Syndrome associated leukemia[26], single-cell derived colonies were analyzed where only 1 gRNA was utilized in order to target the 5′ splice site of exon 2 (Supplementary Fig. 6b–d). Similarly, there is an increase in the

number of megakaryocytic lineage positive colonies in GATA1 splice junction edited LT-HSCs (M, Meg) and ST-HSCs and MEPs (M, E, Meg) compared to control-edited colonies. In addition to the engineering of CRISPR/Cas9-mediated isoform re-arrangements, single gRNA mediated knock-outs can also be utilized in our method; for example, they can be used against STAG2 with single cell CRISPR/Cas9 efficiencies as high as 80–90% in LT-HSCs (Supplementary Fig. 6e, f). In summary, purified LT-HSCs and more committed stem and progenitor cells can be edited with high efficiency by CRISPR/Cas9-mediated NHEJ and HDR, and the effects of gene editing on differentiation can be read out reliably using single cell in vitro assays.

**Long-term xenotransplantation of CRISPR/Cas9-edited LT-HSCs.** Human LT-HSC can only be evaluated functionally using the gold standard xenograft assay[35]. To investigate the functional consequences of exclusive GATA1-Short expression in LT-HSCs in vivo, we performed near-clonal xenotransplantation assays in NSG mice. Limiting dilution analysis[36] of CRISPR/Cas9 control-edited LT-HSCs injected into NSG mice for 24 weeks revealed a repopulating stem cell frequency of ~1/100 edited cells (Supplementary Fig. 7a). To achieve near-clonal xenotransplantation, we transplanted control- or GATA1-Short-edited LT-HSCs into NSG mice at an equivalent dose of 100–150 cells/mouse and after 24 weeks analyzed bone marrow (BM) cells harvested from the injected right femur (RF) and the left femur plus both tibias (BM). Only mice with human $CD45^+$ engraftment levels of >5% in the RF and >90% CRISPR/Cas9 editing efficiency as determined by PCR and Sanger sequencing were included in our analysis: 40% of control mice and 35% of mice transplanted with

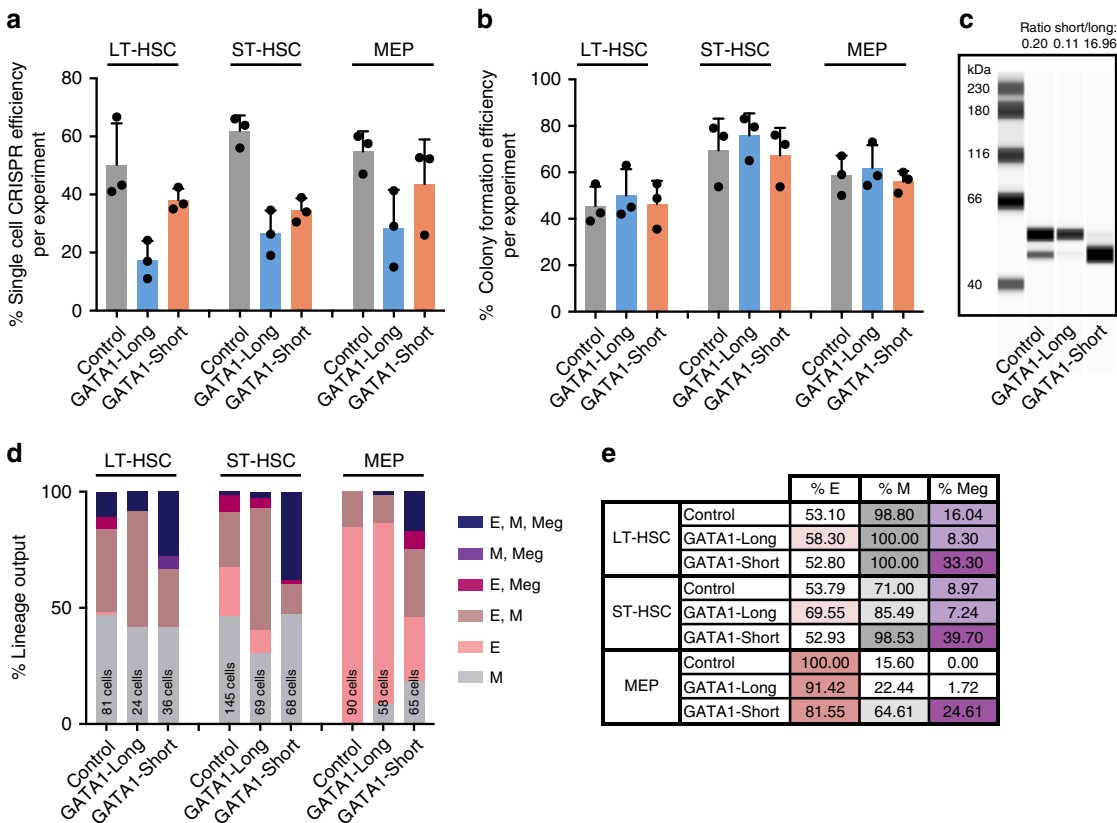

**Fig. 2** Functional interrogation of single CRISPR/Cas9-edited hematopoietic stem cells in vitro. **a** Percentage of CRISPR/Cas9 efficiency as determined by single cell-derived colonies that were positive for homozygous deletion of exon 1 in control OR2W5, positive for exclusive assignment to the GATA1-Long isoform through HDR-mediated mutation of the alternative start site or positive for exclusive assignment to the GATA1-Short isoform through deletion of exon 2 (n = 3 experiments with independent cord blood pools). **b** Percentage of single cells that grew into a colony in the single cell in vitro differentiation assays (n = 3 experiments with independent cord blood pools, unpaired t test P < 0.05 for % colony efficiency in LT-HSCs GATA1-Long versus ST-HSCs GATA1-Long). **c** Capillary-based western assay of GATA1 in bulk MEPs that were CRISPR/Cas9 edited with control, GATA1-Long and GATA1-Short gRNAs and cultured under erythro-myeloid conditions for three days (n = 2 experiments, second western assay is shown in Supplementary Fig. 6a). **d** Lineage output from the in vitro single cell differentiation assay of individual CRISPR/Cas9-edited LT-HSCs, ST-HSCs and MEPs with control, GATA1-Long and GATA1-Short gRNAs. Numbers of single cell colonies with positive genotype are indicated at each condition (E = erythroid, M = myeloid, Meg = megakaryocytic, n = 3 experiments with independent cord blood pools, number of single cells assessed in each condition is indicated in each bar graph, unpaired t test P < 0.01 for E, M, Meg in GATA1-Short versus control and E, M, Meg in GATA1-Short versus GATA1-Long among all three cell types). **e** Overall percentage of erythroid, myeloid and megakaryocyte containing colonies from single cell in vitro differentiation assays (E = erythroid, M = myeloid, Meg = megakaryocytic, n = 3 experiments with independent cord blood pools, unpaired t test P < 0.005 for % Meg in GATA1-Short versus Control or % Meg in GATA1-Short versus GATA1-Long among all three cell types). Error bars represent standard deviations

GATA1-Short edited LT-HSCs fulfilled these criteria (Fig. 3a, Supplementary Fig. 7b–f). To precisely determine the genotype of the clonal progeny of injected LT-HSCs, secondary methylcellulose colony formation assays from cells of the RF were carried out. Analysis of CRISPR/Cas9 edits in individual colonies by Sanger sequencing revealed clonal engraftment in 3 out of 5 GATA1-Short edited LT-HSCs injected mice, highlighting that the xenotransplantations were indeed performed at near clonal levels (Supplementary Fig. 7g). On average, human CD45[+] engraftment in the RF was 40%, both in control and GATA1-Short edited LT-HSCs injected mice (Fig. 3b). Interestingly, GATA1-Short edited LT-HSCs generated grafts with twofold higher percentage of human CD41[+]CD45[−] megakaryocytic lineage derived cells in the RF compared to controls (Fig. 3c). In addition, we observed an increase in the percentage of human CD19[+]CD45[+] B-lymphoid cells in the RF (Fig. 3d, Supplementary Fig. 8a–d). GATA1-Short edited LT-HSCs generated grafts with higher absolute cell numbers, mainly due to increased numbers of B-lymphoid lineage cells (Fig. 3e, Supplementary Fig. 8e–i). The observed increase in megakaryocytic lineage

output in mice transplanted with GATA1-Short edited LT-HSCs (Fig. 3f) confirms our single cell in vitro findings, and demonstrates the feasibility of conducting near-clonal xenotransplantation assays using purified CRISPR/Cas9-edited LT-HSCs.

Normal human HSCs show a predominant B-lymphoid bias upon long-term engraftment in NSG mice[37]. We therefore repeated our xenotransplantation assays using c-kit-deficient NSGW41 recipients, which support enhanced erythropoietic and megakaryocytic lineage output[38]. Limiting dilution analysis of CRISPR/Cas9-edited LT-HSCs in NSGW41 mice for 12 weeks revealed a repopulating cell frequency of ~1/175 (Supplementary Fig. 9a). Consequently, an equivalent dose of 200–250 control- or GATA1-Short-edited LT-HSCs were transplanted into NSGW41 mice and human engraftment in the RF and BM was analyzed after 12 weeks. Totally, 35% of mice transplanted with control edited LT-HSCs and 30% of mice transplanted with GATA1-Short edited LT-HSCs showed robust engraftment and successful CRISPR/Cas9 editing (Fig. 4a, Supplementary Fig. 9b–d). Human CD45[+] engraftment was observed at comparable high levels as in NSG mice (Fig. 4b). Strikingly, a threefold increase in CD41[+]

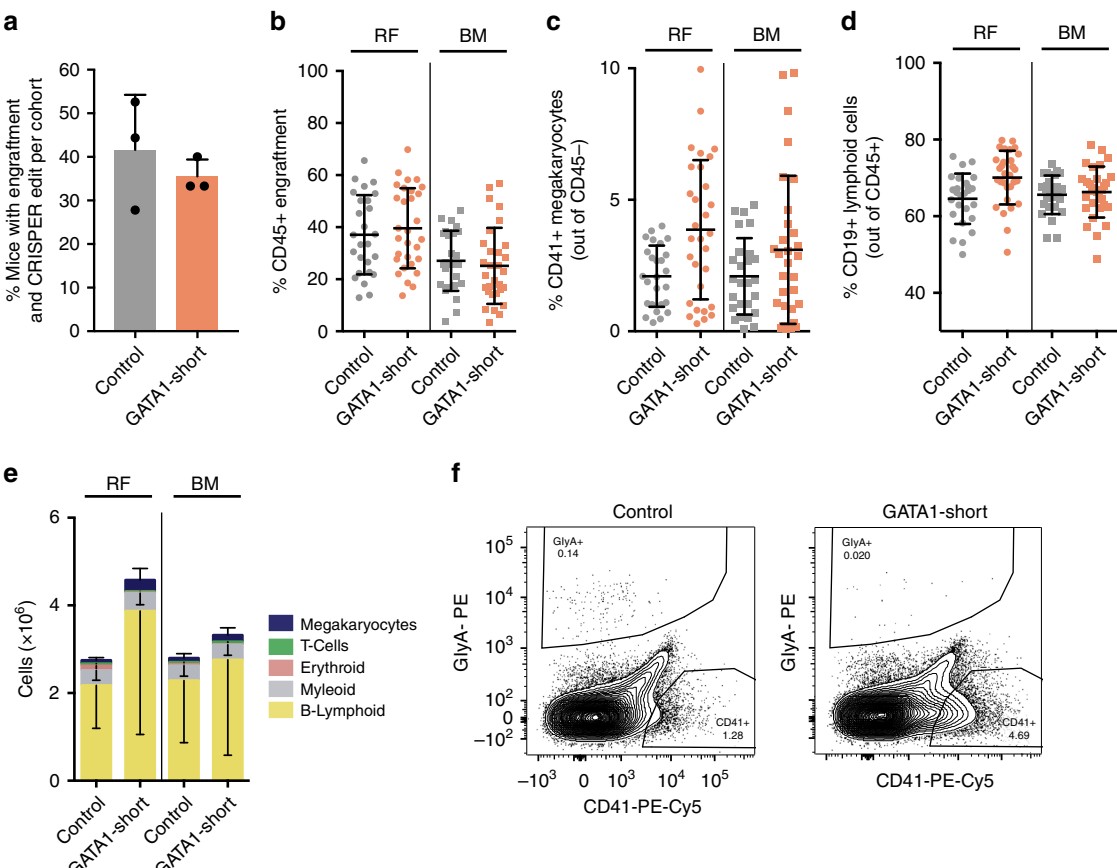

**Fig. 3** Functional interrogation of single CRISPR/Cas9-edited hematopoietic stem cells in NSG mice. **a** Percentage of CRISPR/Cas9 edited LT-HSCs injected NSG mice with engraftment (>5% based on human CD45$^+$ expression in RF) and high CRISPR/Cas9 knockout efficiency (>90% based on PCR and Sanger sequencing) from each of three cohorts ($n = 3$ animal cohorts with independent cord blood pools). **b** Engraftment levels of control and GATA1-short edited LT-HSCs injected NSG mice based on human CD45$^+$ expression in RF and BM. **c** Percentage of CD41$^+$CD45$^-$ megakaryocytes of control and GATA1-Short edited LT-HSCs injected NSG mice in RF and BM (unpaired $t$ test $P < 0.005$ for RF GATA1-Short versus RF control and unpaired $t$ test $P = 0.107$ for BM GATA1-Short versus BM control). **d** Percentage of CD19$^+$CD45$^+$ lymphoid cells in control and GATA1-Short edited LT-HSCs injected NSG mice in RF and BM (unpaired $t$ test $P < 0.005$ for RF GATA1-Short versus RF control). **e** Absolute cell numbers of different lineages in control and GATA1-Short edited LT-HSCs injected NSG mice in RF and BM (unpaired $t$ test $P < 0.01$ for B-lymphoid RF GATA1-Short versus RF control and unpaired $t$ test $P < 0.005$ for megakaryocytes RF GATA1-Short versus RF control). **f** Representative flow cytometry plots of control and GATA1-Short edited LT-HSCs injected NSG mice in RF showing CD41$^+$CD45$^-$ megakaryocytes and GlyA$^+$CD45$^-$ erythroid cells. Error bars represent standard deviations

megakaryocytic lineage output was observed in the RF and BM of mice transplanted with GATA1-Short edited LT-HSCs (Fig. 4c). No changes were seen in the percentage of B-lymphoid cells, but a decrease in human GlyA$^+$CD45$^-$ erythroid cells was detected in mice transplanted with GATA1-Short edited LT-HSCs (Fig. 4d, Supplementary Fig. 10a, b). Analysis of total cell numbers revealed a similar pattern, including a threefold increase in the number of megakaryocytic lineage derived cells and, at the same time, a sixfold reduction in erythroid cell numbers (Fig. 4e, Supplementary Fig. 10c–g). The cellular morphology of GlyA$^+$CD45$^-$ cells revealed more immature forms of erythroid cells and fewer enucleated erythroblasts in the RF of GATA1-Short edited LT-HSCs injected NSGW41 mice compared to control (Supplementary Fig. 11a, b)[39]. No difference in morphology was observed in CD41$^+$ megakaryoblast-like cells in the BM of GATA1-Short edited LT-HSCs injected NSGW41 mice compared to control (Supplementary Fig. 11c). Only bulk CD41$^+$ cells could be sorted, regardless of CD45 staining, since sufficient numbers of CD41$^+$CD45$^-$ megakaryocytes could not be detected after the freeze/thaw cycle of stored BM. Furthermore, there were no differences in B-lymphoid proliferation in grafts generated by control- and GATA1-Short edited LT-HSCs in NSGW41 mice, in

contrast to our observations in NSG recipients, which was possibly due to the reduced lymphoid bias observed in NSGW41 mice. In summary, xenotransplantation into NSGW41 mice further augmented the megakaryocytic lineage output of GATA1-Short edited LT-HSCs in vivo, with a concomitant decrease in erythroid cells (Fig. 4f).

## Discussion

We demonstrate that distinct numbers of isolated LT-HSCs as well as more committed stem and progenitor cells can be edited with high efficiency using CRISPR/Cas9. Conventional CRISPR/Cas9 approaches on bulk CD34$^+$ populations yield a mixed population of edited and unedited cells, making functional characterization difficult. Our approach, which requires retrospective verification of the CRISPR/Cas9 edits, permits the functional interrogation of LT-HSCs at a single cell level. Alternative approaches utilize HDR-mediated stable integration of selectable markers that allow prospective enrichment of CRISPR/Cas9 edited cells upon expression of the fluorescent marker[10,11]. The advantages of our method are that no exogenous sequences are introduced into the genomic DNA and all regulatory processes such as splicing and spatial control of promoters and

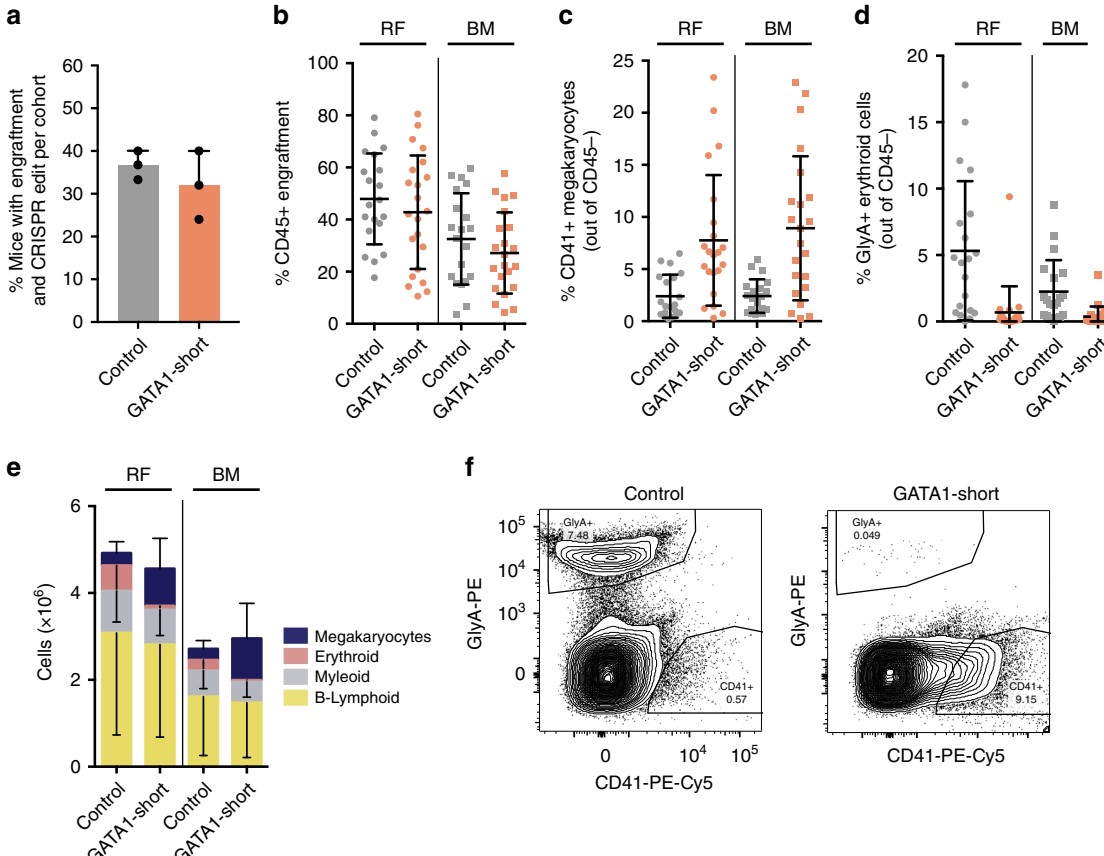

**Fig. 4** Functional interrogation of single CRISPR/Cas9-edited hematopoietic stem cells in NSGW41 mice. **a** Percentage of CRISPR/Cas9 edited LT-HSCs injected NSGW41 mice with engraftment (>5% based on human CD45[+] expression in RF) and high CRISPR/Cas9 knockout efficiency (>90% based on PCR and Sanger sequencing) from each of three cohorts ($n = 3$ animal cohorts with independent cord blood pools). **b** Engraftment levels of control and GATA1-short edited LT-HSCs injected NSGW41 mice based on human CD45[+] expression in RF and BM. **c** Percentage of CD41[+]CD45[−] megakaryocytes of control and GATA1-Short edited LT-HSCs injected NSGW41 mice in RF and BM (unpaired t test $P < 0.001$ for GATA1-Short versus control for both RF and BM). **d** Percentage of GlyA[+]D45[−] erythroid cells in control and GATA1-Short edited LT-HSCs injected NSGW41 mice in RF and BM (unpaired t test $P < 0.0005$ for RF GATA1-Short versus RF control and unpaired t test $P < 0.005$ for BM GATA1-Short versus BM control). **e** Total cell numbers of different lineages in control and GATA1-Short edited LT-HSCs injected NSGW41 mice in RF and BM (unpaired t test $P < 0.005$ for megakaryocytes GATA1-Short versus control for both RF and BM and unpaired t test $P < 0.005$ for erythroid GATA1-Short versus control for both RF and BM). **f** Representative flow cytometry plots of control and GATA1-Short edited LT-HSCs injected NSGW41 mice in RF showing CD41[+]CD45[−] megakaryocytes and GlyA[+]CD45[−] erythroid cells. Error bars represent standard deviations

enhancers are kept intact. In addition, we show for the first time that GATA1-Short isoform expression in LT-HSCs under the endogenous promoter leads to an increase in megakaryocytic output; this was not previously observed in cord blood derived bulk CD34[+] HSPCs[31]. Aside from showing how a prototypical transcription factor functions in regulating lineage fate in LT-HSCs, this result is of great importance in modeling the normal functions of the GATA1 transcription factor and the hematological diseases where it is mutated.

Interestingly, our in vitro and in vivo data seem to closely phenocopy several murine studies of GATA1 loss[40], where megakaryocytes fail to undergo terminal differentiation and these immature megakaryocytes expand dramatically in the BM and spleen. Loss of GATA1 in mouse embryo-derived stem cells results in a maturation arrest at the proerythroblast stage of definitive erythroid precursors[41] and ablation of GATA1 in adult mice results in a maturation arrest at the proerythroblast stage[42]. However, direct comparison of murine and human data is complicated by the fact that human and mouse GATA1-Short are normally produced by divergent mechanisms, where human GATA1-Short is generated through differential splicing and murine GATA1-Short is produced by alternative translation of a single mRNA[43]. Indeed,

GATA1-Short mRNA transcripts have not been reported in murine tissues[43]. Furthermore, although fetal hematopoiesis is perturbed in mice with mutations conferring GATA1-Short expression, adult mice display no obvious hematopoietic defects[32].

Children born with Down syndrome have an increased risk of developing AMKL, which is preceded by a pre-leukaemic syndrome termed transient leukemia that is characterized by high numbers of abnormal megakaryoblasts in the circulation, spleen and liver[44]. Nearly all cases of transient leukemia and Down syndrome associated AMKL have N-terminal truncating GATA1 mutations (GATA1-Short) present at birth that become undetectable after transient leukemia and AMKL remission[28,45,46]. Notably, several transgenic mouse models of trisomy 21 and GATA1-Short have been generated, yet none fully recapitulate the hematological abnormalities and malignancies seen in human trisomy 21[32,47–49]. Thus, the establishment of humanized in vivo model systems suitable to study the pathogenesis of Down syndrome blood malignancy is imperative. While our work described here represents an initial step, we foresee CRISPR/Cas9-mediated GATA1-Short editing in primary trisomy 21 LT-HSCs in combination with near clonal xenotransplantation as the most suitable approach to generate these models.

Our observed phenotype of increased megakaryopoiesis due to forced expression of GATA1-Short seems to be at odds with a subset of Diamond Blackfan anemia patients, which is a genetic disease that usually presents in infancy and is characterized by low red blood cell counts. Most patients with Diamond Blackfan anemia harbor mutations in genes coding for ribosomal subunits, the most common being RPS19, that lead to a selective impaired production of full-length GATA1 and GATA1-Short[23,50,51]. Germline GATA1 mutations have also recently been described in patients with Diamond Blackfan anemia[22–24], which lead to almost exclusive GATA1-Short expression, but no observed increase in megakaryocytes. Thus, it is possible that in these particular cases, germline GATA1 mutations severely constrained fetal blood development inducing an unidentified compensatory mechanism to allow for a more normalized megakaryocytic development. This selective pressure is further supported by the fact that mothers of children with Diamond Blackfan anemia have increased numbers of miscarriages[52].

In summary, our method opens up the possibility of studying gene function relationships not only in LT-HSCs, but also in other stem and progenitor cells to uncover cell type specific phenotypes. We believe that the continuous improvement of CRISPR/Cas9 editing efficiency, for example through chemically modified gRNAs[53–55] or different Cas9 variants[56,57], will further improve this approach. In the future, we envision that this method could potentially be adapted for cellular therapies.

## Methods

**Cord blood lineage depletion.** Human cord blood samples were obtained from Trillium and William Osler hospitals with informed consent in accordance to guidelines approved by University Health Network (UHN) Research Ethics Board. Cord blood samples were processed 24–48 h after birth. Male samples were exclusively utilized in this study because GATA1 and STAG2 are located on the X-chromosome. This aided the CRISPR/Cas9 efficiency because of the need to edit only one X-chromosome. Control OR2W5 is located on chromosome 1. Samples were diluted 1:1 with phosphate-buffered saline (PBS) and mononuclear cells were enriched using lymphocyte separation medium (Wisent, 305-010-CL). Subsequently, red blood cells were lysed using an ammonium chloride solution (Stem-Cell Technologies, 07850). Then, lineage positive cells were depleted by negative selection with the StemSep Human Hematopoietic Progenitor Cell Enrichment Kit (StemCell Technologies, 14056) and Anti-Human CD41 TAC (StemCell Technologies, 14050) according to the manufacturer's protocol. Lineage depleted cells were stored in 50% PBS, 40% fetal bovine serum (FBS) (ThermoFisher, 12483-020) and 10% DMSO (FisherScientific, D128-500) at −150 °C.

**Cord blood sorting.** Lineage depleted cells were thawed via slow dropwise addition of X-VIVO 10 (Lonza, 04743Q) with 50% FBS (Sigma, 15A085) and DNaseI (200 μg/ml, Roche, 10104159001). Cells were spun at 350×g for 10 min at 4 °C and then resuspended in PBS + 2.5% FBS. For all in vitro and in vivo experiments, the full stem and progenitor hierarchy sort as described in Notta et al.[34] was utilized in order to sort LT-HSCs, ST-HSCs, and MEPs. Lineage depleted cells were resuspended in 100 μl per 1 × 10⁶ cells and stained in two subsequent rounds for 20 min at room temperature each. First, the following antibodies were used (volume per 1 × 10⁶ cells, all from BD Biosciences, unless stated otherwise): CD45RA FITC (5 μl, 555488, HI100), CD49f PE-Cy5 (3.5 μl, 551129, GoH3), CD10 BV421 (4 μl, 562902, HI10a), CD19 V450 (4 μl, 560353, HIB19), and FLT3 CD135 biotin (12 μl, clone 4G8, custom conjugation). After washing the cells, a second set of antibodies was used (volume per 1 × 10⁶ cells, all from BD Biosciences, unless stated otherwise): CD45 V500 (4 μl, 560777, HI30), CD34 APC-Cy7 (3 μl, clone 581, custom conjugation), CD38 PE-Cy7 (2.5 μl, 335825, HB7), CD90 APC (4 μl, 559869, 5E10), CD7 A700 (10 μl, 561603, M-T701), and Streptavidin Conjugate Qdot 605 (3 μl, ThermoFisher, Q10101MP). Cell sorting was performed on the FACSAria III (BD Biosciences). LT-HSCs were sorted as CD45⁺CD34⁺CD38⁻CD45RA⁻CD90⁺CD49f⁺, ST-HSCs as CD45⁺CD34⁺CD38⁻CD45RA⁻CD90⁻CD49f⁻ and MEPs as CD45⁺CD34⁺CD38⁺CD10/19⁻CD7⁻CD45RA⁻FLT3⁻ (Supplementary Figs. 1 and 2).

**Pre-electroporation culture of sorted cells.** Sorted LT-HSCs, ST-HSCs or MEPs were cultured for 36–48 h in serum-free X-VIVO 10 (Lonza) media with 1% Bovine Serum Albumin Fraction V (Roche, 10735086001), 1× L-Glutamine (Thermo Fisher, 25030081), 1× Penicillin–Streptomycin (Thermo Fisher, 15140122) and the following cytokines (all from Miltenyi Biotec): FLT3L (100 ng/mL), G-CSF (10 ng/mL), SCF (100 ng/mL), TPO (15 ng/mL), and IL-6 (10 ng/mL). Cells were cultured in 96-well U-bottom plates (Corning, 351177).

**gRNA and HDR template design.** gRNAs for GATA1 Short and Long were designed on Benchling (http://www.benchling.com). For GATA1 Short, gRNAs sequences were considered that were flanking the 5′ and 3′ end of exon 2. Individual gRNAs targeting the 5′ or 3′ end were individually tested for cleavage efficiency and the best gRNA targeting each end was selected. Combined use of both gRNAs enabled complete excision of exon 2 (Fig. 1b). For GATA1 Long, gRNA sequences closest to the second ATG start codon were individually tested for cleavage efficiency and the best gRNA was selected. The GATA1 Long HDR template was designed with 60 bp homology ends at either side. For the template, the ATG (Methionine) start codon was mutated to CTC (Leucine) and the PAM sequence was mutated from GGG (Glycine) to GGC (Glycine) in order to avoid repeated cutting by the gRNA (Fig. 1c). The control gRNAs, which target exon 1 of the olfactory receptor OR2W5, were predicted by the CRoatan algrotihm[33]. The STAG2 gRNA was predicted with the same algorithm.

gRNA and HDR template sequences:
Control gRNA-1: GACAACCAGGAGGACGCACT
Control gRNA-2: CTCCCGGTGTGGACGTCGCA
GATA1 Short gRNA-1: TGGAACGGGGAGATGCAGGA
GATA1 Short gRNA-2: CCACTCAATGGAGTTACCTG
GATA1 Long gRNA: CATTGCTCAACTGTATGGAG
GATA1 Long HDR template:
TCTTTCCTCCATCCCTACCTGCCCCCAACAGTCTTTCAGGTGTACCCATTGCTCAACTGTCTCGAGGGCATCCCAGGGGGCTCACCATATGCCGGCTGGGCCTACGGCAAGACGGGGCTCTACCCTGCC
STAG2 gRNA: AATGGTCATCACCAACAGAA

**CRISPR/Cas9 RNP electroporation.** gRNAs were synthesized from IDT as Alt-R CRISPR/Cas9 crRNA, which require annealing with Alt-R tracrRNA (IDT) to form a functional gRNA duplex. The HDR template was synthesized from IDT as a single-strand Ultramer. crRNAs and tracrRNAs were resuspended to 200 μM with TE Buffer (IDT). Both RNA oligonucleotides were mixed 1:1 to a final concentration of 100 μM and annealed at 95 °C for 5 min in a thermocycler, then cooled down to room temperature on the bench top. If using two gRNAs at the same time, both crRNAs were annealed to the tracrRNA in a single tube. For each reaction, 1.2 μl crRNA:tracrRNA, 1.7 μl Cas9 protein (IDT) and 2.1 μl PBS were combined in a low-binding Eppendorf tube (Axygen, MCT-175-C-S) and incubated for 15 min at room temperature. Subsequently, 1 μl of 100 μM electroporation enhancer (IDT) was added. Pre-electroporation cultured cells were washed in warm PBS and spun down at 350×g for 10 min at room temperature. Between 1 × 10⁴–5 × 10⁴ cells per condition were resuspended in 20 μl of Buffer P3 (Lonza) per reaction and added to the Eppendorf tube containing the Cas9 gRNA RNP complex. The mixture was briefly mixed by pipetting and then added to the electroporation chamber (Lonza, V4XP3032). Cells were electroporated with the program DZ-100 using the Lonza Nucleofector and, immediately afterwards, 180 μl of pre-warmed X-VIVO 10 media (as described above) was added. Cells were recovered overnight in the incubator before their use in in vivo or single cell in vitro assays.

**Setup of single cell in vitro assay.** Single cell in vitro assays were performed according to Notta et al.[34] with slight modifications. Briefly, 3 days prior to the single cell assay, Nunc 96-well flat bottom plates (Thermo Fisher, 167008) were coated with 50 μl per well of 0.2% Gelatin (G1393, Sigma-Aldrich) using a multichannel pipette. The first and last column of each 96-well plate were not utilized and instead filled with 150 μl PBS per well to support proper humidity within the plate. After 1 h, the 0.2% Gelatin is removed and murine MS-5 stroma cells[58] were seeded at a density of 1500 cells per well in 100 μl H5100 media (StemCell Technologies, 05150). One-day prior, the H5100 media was removed and replaced with 100 μl of erythro-myeloid differentiation media. For this, StemPro-34 SFM media (Thermo Fisher, 10639011) is used with the supplied supplement, 1× L-Glutamine (Thermo Fisher, 25030081), 1× Penicillin–Streptomycin (Thermo Fisher, 15140122), 10 μl Human LDL (StemCell Technologies, 02698) for every 50 ml media and the following cytokines (all from Miltenyi Biotec unless stated otherwise): FLT3L (20 ng/mL), GM-CSF (20 ng/mL), SCF (100 ng/mL), TPO (100 ng/mL), EPO (3 ng/mL, Eprex), IL-2 (10 ng/mL), IL-3 (10 ng/mL), IL-6 (50 ng/mL), IL-7 (20 ng/mL), and IL-11 (50 ng/mL). On the day of the experiment, electroporated cells were stained with 1:2000 Sytox Blue (ThermoFisher, S34857) in PBS + 2.5% FBS and viable single cells were sorted and deposited into the MS-5 seeded 96 well plate using the FACSAria II (BD Biosciences). Single cells were cultured for 16–17 days with an addition of 100 μl erythro-myeloid differentiation media per well at day 8.

**Flow cytometry of single cell in vitro assay.** Wells with hematopoietic cell content were marked 1 day prior and the total number of wells with colonies was used to calculate CRISPR/Cas9 and single cell colony efficiencies. On the day of analysis, 140 μl of media was removed from each well with a multichannel pipette and the content was mixed well. Upon additional washing of the wells with PBS, the content in each well was transferred to a 96-well filter plate (8027, Pall) in order to remove stromal cells. For this, the filter plate was put on top of a 96-well U-bottom plate (Corning, 351177) and centrifuged at 300×g for 7 min at room

temperature. After the cells were pelleted at the bottom of the U-bottom plate, the media was tossed by quickly inverting the U-bottom plate (~20 μl of liquid per well remained). Then, 30 μl of PBS + 2.5% FBS was added and mixed into each well with a multichannel pipette and 25 μl (half) was transferred to a 96-well PCR plate (Eppendorf, 951020362) for subsequent genomic DNA extraction (the PCR plate was sealed and stored at -80C). After that, 25 μl of antibody mix in PBS + 2.5% FBS was added to each well, containing 25 μl of hematopoietic cells, and stained for 45 min at 4 °C. The following antibodies were used (all antibodies from BD Biosciences, unless stated otherwise): CD45 APC (1:100, 560777, HI30), CD34 APC-Cy7 (1:250, clone 581, custom conjugation), CD33 BV421 (1:50, Biolegend, 303416, WM53), CD71 FITC (1:100, 347513, L01.1), CD41 PE-Cy5 (1:200, Beckman Coulter, 6607116, P2) and GlyA PE (1:200, Beckman Coulter, IM2211U, KC16). Finally, 150 μl of PBS + 2.5% FBS were added to each well with a multichannel pipette and the cells were analyzed on the FACSCelesta with a high throughput sampler (HTS, BD Biosciences). All flow cytometry quantification was performed in a blinded manner. Generally, greater than ten cells were required to call a positive lineage. Erythroid cells were defined as positive upon CD71 expression, with or without expression of GlyA.

**Genotyping of single cell in vitro assay.** The PCR plates containing 25 μl of cells per well were thawed and a modified protocol of the Agencourt GenFind V2 (Beckman Coulter, A41499) was utilized to isolate genomic DNA. Wells with cell content were transferred to a new 96-well PCR plate (Eppendorf, 951020362) in order to utilize multichannel pipetting for each future step. Totally, 25 μl of lysis buffer and 1.2 μl of Proteinase K (Zymo Research, D3001220) were pipetted into each well. After 30 min at room temperature, 50 μl of magnetic particles were added. After 5 min, the PCR plate was placed on a magnetic stand (ThermoFisher, AM10027) for 10 min. The supernatant was removed with a multichannel pipette and the plate was taken off the magnet. Totally, 200 μl of wash buffer 1 were mixed into each well and the PCR plate was put back onto the magnetic stand. After 10 min, the supernatant was removed and each well was washed with 125 μl of wash buffer 2. Finally, after the last wash buffer was removed, the magnetic particles were resuspended with 60 μl of TE buffer (IDT). The PCR plate was put back on the magnetic stand and after 10 min, 57 μl of eluted genomic DNA was removed and put into a new PCR plate.

The CRISPR/Cas9 engineered genomic locus was amplified via PCR. For each PCR reaction, 23 μl of eluted genomic DNA was mixed with 1 μl of forward and reverse primer (10 μM) and 25 μl of AmpliTaq Gold 360 Master Mix (ThermoFisher, 4398881). The PCR program was: 95 °C for 10 min, followed by 95 °C for 30 s, 56 °C for 30 s and 72 °C for 1 min (40 cycles) and then 72 °C for 7 min. To identify colonies with the GATA1-Short genotype, 15 μl of PCR product was run on a 1.5% agarose gel (ThermoFisher, 16500500). Control gRNA colonies were screened for homozygous deletion of OR2W5 (deletion within exon 1, 700 to 500 bp, (Supplementary Fig. 4a). Similarly, GATA1-Short colonies were screened for a shift in the size of the PCR product (deletion of exon 2, 1000 to 550 bp, Supplementary Fig. 4b). In order to identify colonies with the GATA1-Long genotype, PCR products were column purified using the ZR-96 DNA Clean-up Kit (Zymo Research, D4018) according to the manufacturer's protocol. The purified PCR product was Sanger sequenced using the reverse PCR primer and the chromatograms were inspected to identify colonies that contained the alternative start site mutation (Supplementary Fig. 4c). Finally, to identify STAG2 knock-out colonies, PCR products were column purified and sent for Sanger sequencing using the reverse PCR primer. Only colonies that showed a frame shift mutation were considered positive (Supplementary Fig. 6f).

PCR primers:
Control (OR2W5) forward primer: 5′-TCGGCCTGGACTGGAGAAAA-3′
Control (OR2W5) reverse primer: 5′-GAGACCACTGTGAGGTGAGA-3′
GATA1-Short forward primer: 5′-CAGGAGAGAATGAGAAAAGAGTGGA-3′
GATA1-Short reverse primer: 5′-ATTTCCAAGTGGGTTTTTGAGGAT-3′
GATA1-Long forward primer: 5′-GCCACACTGAGAGGCAATACT-3′
GATA1-Long reverse primer: 5′-AAAAGTCAGGGCCCCCATAAG-3′
STAG2-forward primer: 5′-CCACAAAGAGGCTGTCACAGTT-3′
STAG2-reverse primer: 5′-CATGCAGCAGAAAATGAATCAAAAC-3′

**Western assay.** MEPs were sorted and CRISPR/Cas9 RNP electroporated as described above. After electroporation, MEPs were cultured in erythro-myeloid differentiation media (as described above for single cells in vitro assays) for 3 days. 1 × 10^5 cells were then lysed in RIPA buffer (ThermoFisher, 899000) containing protease and phosphatase inhibitors (ThermoFisher, 78446). Subsequently, samples were centrifuged at 12,000 g for 5 min at 4 °C and supernatants were utilized for Western assay. Western assay was performed on the size-based Wes capillary platform (ProteinSimple) using the 12–230 kDa capillary cartridge according to manufacturer's protocol. In order to determine the optimal antibody dilution, anti-GATA-1 antibody (Cell Signaling D24E4) was titrated prior to use on a lysate from CD34+ cord blood and 1:5 dilution was used in the samples.

**Animal studies.** All mouse experiments were approved by the University Health Network (UHN) Animal Care Committee and we complied with all relevant ethical regulations for animal testing and research. All mouse transplants were performed with 8- to 12-week-old female *NOD.Cg-Prkdc^scidIl2rg^tm1Wjl/SzJ* (NSG) mice (JAX) that were sublethally irradiated with 225 cGy, 24 h before transplantation, or with 8- to 12-week-old female *NOD.Cg-Prkdc^scidIl2rg^tm1WjlKit^em1Mvw/SzJ* (NSGW41) mice that were not irradiated. Sample size was chosen to give sufficient power for calling significance with standard statistical tests. Intrafemoral injections were performed as described in Mazurier et al[59]. For this, mice were anesthetized with isoflurane and the right knee was secured in a bent position to drill a hole into the RF with a 27 gauge needle. Then, 100–250 CRISPR/Cas9-edited LT-HSCs were injected in 30 μl PBS using a 28 gauge ½ cc syringe (Becton Dickinson, 329461). LT-HSC cell numbers are based on the number of flow cytometry sorted cells at day 0. After 12 or 24 weeks, mice were sacrificed to obtain the RF and BM (left femur and both tibias, BM). Bones were flushed in 1 mL PBS + 2.5% FBS and cells were centrifuged at 350×g for 10 min. Cells were resuspended in 500 μl of PBS + 2.5% FBS. Subsequently, cells from BM and RF were counted in ammonium chloride (StemCell Technologies, 07850) using the Vicell XR (Beckman Coulter). Totally, 25 μl of cells were used for flow cytometry analysis and another 25 μl were frozen down for genomic DNA isolation in order to verify CRISPR/Cas9 edits (same protocol as above for single cell in vitro assays).

**Limiting dilution in vivo assays.** For limiting dilution transplantation assays, CRISPR/Cas9 control gRNA electroporated LT-HSCs were injected at defined doses (equivalent to 25, 50, 100, and 200 LT-HSCs) into 8- to 12-week-old female NSG or NSGW41 mice. LT-HSC cell numbers were based on the number of flow cytometry sorted cells at day 0. LT-HSC frequency was estimated using the online tool ELDA (http://bioinf.wehi.edu.au/software/elda/index.html)[36].

**Flow cytometry of in vivo assays.** Cells from the RF and BM of transplanted mice were stained in 96-well U-bottom plates (Corning, 351177) for 45 min at 4 °C. The following antibodies were used (all from BD Biosciences, unless stated otherwise): CD45 APC-Cy7 (1:100, 348795560566), CD45 A700 (1:100, 560566), CD33 APC (1:100, 340474), and CD19 V450 (1:100, 560353), CD41 PE-Cy5 (1:200, Beckman Coulter, 6607116), GlyA PE (1:100, Beckman Coulter, IM2211U), and CD3 FITC (1:100, 349201). For NSGW41 mice at 12 weeks, CD3 FITC staining was omitted. Cells were analyzed on the FACSCelesta (BD Biosciences).

**CRISPR/Cas9 efficiency in cells and engrafted mice.** After each CRISPR/Cas9 RNP electroporation, a small subset of cells was cultured in X-VIVO 10 media (as described above) for 5–7 days in order to validate CRISPR/Cas9 efficiency. Genomic DNA was isolated from bulk cells and the CRISPR/Cas9 engineered genomic locus was amplified via PCR as described above. Sanger sequencing was carried out using the reverse PCR primer and the chromatograms were analyzed using the online tool TIDE (https://tide.deskgen.com/)[60] in order to verify CRISPR/Cas9 editing in control, GATA1-Short and GATA1-Long edited bulk cells. Because of the large deletion size of control (200 bp) and GATA1-Short (400 bp) edited cells, the CRISPR/Cas9 efficiency was evaluated based on the percentage of aberrant sequences after the gRNA cut site (Supplementary Fig. 7e).

For each transplanted mouse, CRISPR/Cas9 efficiency of control and GATA1-Short edited cells was evaluated in the RF using the same approach. Only mice that showed a CRISPR/Cas9 knockout efficiency of >90% as determined by the percentage of aberrant sequences after the gRNA cut site and a CD45+ engraftment level in the RF of >5% were utilized in the analysis. Because only one X-chromosome needed to be edited for GATA1-Short, single clonal engraftment was visible based on individual chromatograms (Supplementary Fig. 7f). GATA1-Short edited LT-HSCs transplanted mice with more than one clone were included into our near-clonal xenotransplantation analysis, as long as the CRISPR/Cas9 knockout efficiency of >90% and engraftment criteria of >5% were satisfied.

**Methylcellulose colony formation assay.** Totally, 1 × 10^5 cells from the RF of xenotransplanted mice were transferred to 1 ml of MethoCult H4034 Optimum methylcellulose medium (StemCell Technologies, 04034) and plated onto a 35 mm dish for human-specific colony formation. After 10–11 days, individual colonies were collected, washed in PBS and genomic DNA was isolated as described above. Individual CRISPR/Cas9 edits were determined using PCR amplification and Sanger sequencing with the reverse PCR primer as described above.

**Cellular morphology.** The RF and BM of genetically verified control and GATA1-Short edited LT-HSCs injected NSGW41 mice were thawed and five samples per condition were pooled together. Murine cells were depleted from each pooled sample using the Mouse Depletion Kit (Miltenyi Biotec). Cells were stained for 30 min at 4 °C with the following antibodies: CD45 V500 (1:100, 560777, HI30), GlyA PE (1:100, Beckman Coulter, IM2211U), and CD41 PE-Cy5 (1:200, Beckman Coulter, 6607116, P2). GlyA+CD45− sorted cells from RF were washed in PBS with 2.5% FBS and then cytospinned using the CytoSpin 4 instrument (112×g for 10 min, high acceleration, ThermoFisher) according to manufacturer's instructions. CD41+ sorted cells from BM were cytospinned at 55×g for 10 min with medium acceleration.

**CRISPR/Cas9 off-target analysis**. Genomic loci that were similar to the gRNA target sequence were identified with Cas0-OFFinder (http://www.rgenome.net/cas-offinder)[61], using a mismatch number of 2–4 and a DNA/RNA bulge size of 0. Two genomic loci with 2 mismatches and 10 genomic loci with 3 mismatches were chosen for GATA1-Short gRNA-1, 8 genomic loci with 3 mismatches and 1 genomic loci with 4 mismatches for GATA1-Short gRNA-2, 4 genomic loci with 3 mismatches and 7 genomic loci with 4 mismatches for GATA1-Long gRNA-1, 3 genomic loci with 2 mismatches, 2 genomic loci with 3 mismatches and 6 genomic loci with 4 mismatches for Control gRNA-1, and 1 genomic loci with 3 mismatches and 11 genomic loci with 4 mismatches for Control gRNA-2. PCR primers were designed to amplify 500 bp around these genomic loci. 20–30 single cell colonies that were positively identified with the correct CRISPR/Cas9 edit were selected from each cell type and genotyped for PCR amplification. PCR products were column purified and subsequent Sanger sequencing with both the forward and reverse PCR primer was carried out. TIDE analysis was used to assess any CRISPR/Cas9 cleavage efficiency.

PCR primers:

FW_GATA1Short_gRNA1_chr4_118059061: 5′-CCCTCAAACTTCCCGACTAAGG-3′

RV_GATA1Short_gRNA1_chr4_118059061: 5′-GCTCAGATGTGACTGGGCG-3′

FW_GATA1Short_gRNA1_chr16_35544187: 5′-CGCAGGTTTAGCAAAGTTCCC-3′

RV_GATA1Short_gRNA1_chr16_35544187: 5′-CCAGATGGGTTTCTGCTTTTACGA-3′

FW_GATA1Short_gRNA1_chr22_44756609: 5′-GGGTTTTTCTGGGCTGCAAG-3′

RV_GATA1Short_gRNA1_chr22_44756609: 5′-ACATAATAGTATTGCAGTAATGCGG-3′

FW_GATA1Short_gRNA1_chr19_32673314: 5′-CTCTAGGGCCGCTTCTCAGT-3′

RV_GATA1Short_gRNA1_chr19_32673314: 5′-ACGCTACATAGGGTCGTTCC-3′

FW_GATA1Short_gRNA1_chr15_52109094: 5′-GTTTGGACTGGTCGAGCCT-3′

RV_GATA1Short_gRNA1_chr15_52109094: 5′-GAGTGCGAACCTCTCATCTCTC-3′

FW_GATA1Short_gRNA1_chr10_53712490: 5′-CCAGTCAAGCACCTAGCGTA-3′

RV_GATA1Short_gRNA1_chr10_53712490: 5′-GACTAGAAAGGCCTGAACCTCT-3′

FW_GATA1Short_gRNA1_chr14_72742463: 5′-GGTCCCTTCCAGCACGTAAT-3′

RV_GATA1Short_gRNA1_chr14_72742463: 5′-CCTCCCAATCCGGGAACAAC-3′

FW_GATA1Short_gRNA1_chr16_71816748: 5′-TCTGGGGTACTAGAGGGGAAC-3′

RV_GATA1Short_gRNA1_chr16_71816748: 5′-TCAGGCCAGCATACATCGTG-3′

FW_GATA1Short_gRNA1_chr1_223704955: 5′-CTGAAATGTCGGCCCTGCAA-3′

RV_GATA1Short_gRNA1_chr1_223704955: 5′-CCGAGGATGTGAGTCATGGTGG-3′

FW_GATA1Short_gRNA1_chr17_80332977: 5′-CCTTGGGTCTGCGTGAAGAT-3′

RV_GATA1Short_gRNA1_chr17_80332977: 5′-AGTCACCTCTTAACTCTAACAGAAA-3′

FW_GATA1Short_gRNA1_chr10_20221643: 5′-AGTGCAGTTAGAACAGACCAGC-3′

RV_GATA1Short_gRNA1_chr10_20221643: 5′-GACAGACCTGTCAATCTGCTTTAT-3′

FW_GATA1Short_gRNA1_chr18_8376641: 5′-GAGGACTGGCTGTTTCATCG-3′

RV_GATA1Short_gRNA1_chr18_8376641: 5′-CACTATTCTCAGGAGCGGTCT-3′

FW_GATA1Short_gRNA2_chr5_178537945: 5′-CATGACAGGCGCCTTCTTTACT-3′

RV_GATA1Short_gRNA2_chr5_178537945: 5′-GAACATCTCCCCTTCAGCTCCT-3′

FW_GATA1Short_gRNA2_chr13_113094529: 5′-TAGGTGTTGAACATGACCGTGG-3′

RV_GATA1Short_gRNA2_chr13_113094529: 5′-ACAAAGTGCTGCTGAGACAGAG-3′

FW_GATA1Short_gRNA2_chr21_42868084: 5′-AGCACACAACCTTCAAAAGACG-3′

RV_GATA1Short_gRNA2_chr21_42868084: 5′-GTGGGTGGAATGTCACCTTACT-3′

FW_GATA1Short_gRNA2_chr10_123397080: 5′-AATGTCTGAGAGGCTTCCATTCC-3′

RV_GATA1Short_gRNA2_chr10_123397080: 5′-CAAGATGTGCCTCCTCTCTGTG-3′

FW_GATA1Short_gRNA2_chr8_35663175: 5′-TCACAAGTTTGTTAGAGCGGTG-3′

RV_GATA1Short_gRNA2_chr8_35663175: 5′-TGCTCTCAGCAGTAGATCTGG-3′

FW_GATA1Short_gRNA2_chr8_107174119: 5′-CTCAGCGAGGCACAGAATTG-3′

RV_GATA1Short_gRNA2_chr8_107174119: 5′-AGCGGCTACCCCATAATAGC-3′

FW_GATA1Short_gRNA2_chr12_2085541: 5′-TTAAGGCAATGTCAACCTTTTCCA-3′

RV_GATA1Short_gRNA2_chr12_2085541: 5′-AGACGATGCAGGACCACCTA-3′

FW_GATA1Short_gRNA2_chr3_127784219: 5′-TAACCCCCTTTGGGAGCACA-3′

RV_GATA1Short_gRNA2_chr3_127784219: 5′-GCAGTGGGACGGCTATATCT-3′

FW_GATA1Short_gRNA2_chr7_17679912: 5′-AGTATGCACATCCAGTTGGGG-3′

RV_GATA1Short_gRNA2_chr7_17679912: 5′-CTGCTTGCACTGCTTAACGA-3′

FW_GATA1Long_gRNA_chr8_109403836: 5′-CCCAAATATCAGTTCCATTGGGC-3′

RV_GATA1Long_gRNA_chr8_109403836: 5′-GCAGCTGTTAACAAGAGAGTGC-3′

FW_GATA1Long_gRNA_chr3_77895170: 5′-TTCCTCGCATCACATACGCC-3′

RV_GATA1Long_gRNA_chr3_77895170: 5′-ACCAGATTTCAGCCACTTGGA-3′

FW_GATA1Long_gRNA_chr4_22175316: 5′-AGCGCTTGAATACCTTCAGCAT-3′

RV_GATA1Long_gRNA_chr4_22175316: 5′-GGAGGTGGCACAAGGTAGGTA-3′

FW_GATA1Long_gRNA_chr13_51458879: 5′-AGAATGGTGACCCTGGCTTA-3′

RV_GATA1Long_gRNA_chr13_51458879: 5′-ATTGGCTGGACAGTTTCGC-3′

FW_GATA1Long_gRNA_chr15_62326516: 5′-TAGGCGGGAATGTCAAAGCC-3′

RV_GATA1Long_gRNA_chr15_62326516: 5′-GAGCTGGGACCCACGATAAAA-3′

FW_GATA1Long_gRNA_chr8_81290900: 5′-AACGGGCACTTTAACAGGGAA-3′

RV_GATA1Long_gRNA_chr8_81290900: 5′-AACCTTTGTCAGACCAACTCTGT-3′

FW_GATA1Long_gRNA_chr7_40859472: 5′-AAGCTCGTTTGCTGCTTTACTG-3′

RV_GATA1Long_gRNA_chr7_40859472: 5′-CTCCGCTCTTATGCTCTCTAGC-3′

FW_GATA1Long_gRNA_chr12_20017977: 5′-ATTAGGTCAGTCCAAGCCAACA-3′

RV_GATA1Long_gRNA_chr12_20017977: 5′-TCTCCCTCTTATCAATACCCCCAAC-3′

FW_GATA1Long_gRNA_chr12_94844478: 5′-TTGCTAGATACAGTCCTTGCCTG-3′

RV_GATA1Long_gRNA_chr12_94844478: 5′-CCATTGTCTTGCAGATGTTCTCA-3′

FW_GATA1Long_gRNA_chr2_15662510: 5′-ACCGACATGGGCCTAAGGTA-3′

RV_GATA1Long_gRNA_chr2_15662510: 5′-GGCAGCTTGTCTTGAGAATGG-3′

FW_GATA1Long_gRNA_chr9_76619884: 5′-ATCCCTCTCAAGCAGCCTAAT-3′

RV_GATA1Long_gRNA_chr9_76619884: 5′-ATTTGCTCACAGAGAAATCACACCA-3′

FW_Control_gRNA1_chr7_3081168: 5′-AAATTATCCACATTAATCGGGAAGCC-3′

RV_Control_gRNA1_chr7_3081168: 5′-TTGTTTCAAAATATCACCGTTGGCA-3′

FW_Control_gRNA1_chr7_135691300: 5′-CTTTGAGTCTTTCCTCCCATGC-3′

RV_Control_gRNA1_chr7_135691300: 5′-ACTGGGGGTAGCTCTAGTTGAA-3′

FW_Control_gRNA1_chr3_55088990: 5′-CTGTGTCACTTGGCTCGTCT-3′

RV_Control_gRNA1_chr3_55088990: 5′-CATGTAGTGAGTACCGGGGC-3′

FW_Control_gRNA1_chr5_4882788: 5′-AGCAGCCAACTTGGAAACATAAG-3′

RV_Control_gRNA1_chr5_4882788: 5′-AATCAGGCCTCAGAACCATCTT-3′

FW_Control_gRNA1_chr16_48579049: 5′-TTACCACACTCCACTCCTCTGT-3′

RV_Control_gRNA1_chr16_48579049: 5′-CCTGTGTTCCTGAACCTCCTAAT-3′

FW_Control_gRNA1_chr1_191051674: 5′-ATCCAAGCATACATGACAGAGAT-3′
RV_Control_gRNA1_chr1_191051674: 5′-GTATCTTGTGGGCACAACCTTT-3′
FW_Control_gRNA1_chr12_114950867: 5′-CAATCCTGCCACATTTCTGCATAC-3′
RV_Control_gRNA1_chr12_114950867: 5′-CTTGCCTGTCTCTCAAAGCATC-3′
FW_Control_gRNA1_chr21_34622361: 5′-AGGATTTCAGCTCTAGCAACCA-3′
RV_Control_gRNA1_chr21_34622361: 5′-CCCCATTTTAGACTTATGCCCCC-3′
FW_Control_gRNA1_chr2_30448683: 5′-CGTAATAGCGCTTGTGTTTCAGT-3′
RV_Control_gRNA1_chr2_30448683: 5′-GCTAGATGACTGAGTTGGCAC-3′
FW_Control_gRNA1_chr6_169886852: 5′-AACAGATGATTCTGCAGGCAAAC-3′
RV_Control_gRNA1_chr6_169886852: 5′-GACGATCTTTTAGCTGTCCCCA-3′
FW_Control_gRNA1_chr20_52425093: 5′-CAACCTGTACTCTGGGACGTAG-3′
RV_Control_gRNA1_chr20_52425093: 5′-CGTGGAAACTTCTTCCTTGTCG-3′
FW_Control_gRNA2_chr4_1100146: 5′-TGTAAAGGCGTGCTGTTGGT-3′
RV_Control_gRNA2_chr4_1100146: 5′-ATCTGACGGTGTTGTGACTGG-3′
FW_Control_gRNA2_chr16_85653916: 5′-GTGTGTGTGAAGGCCGGTA-3′
RV_Control_gRNA2_chr16_85653916: 5′-CTCCACAATGAGGCGTTCCC-3′
FW_Control_gRNA2_chr2_118858930: 5′-CAGGAGACGCTCCAAGTCG-3′
RV_Control_gRNA2_chr2_118858930: 5′-TCCCTCACGTACCAGTCGG-3′
FW_Control_gRNA2_chr21_42232307: 5′-ATGGACCAACTCTCACAGCA-3′
RV_Control_gRNA2_chr21_42232307: 5′-GGCCTCCATTTGAGAACGTG-3′
FW_Control_gRNA2_chr12_12496109: 5′-TGTGGCGTGTAGGTAAAGAGAC-3′
RV_Control_gRNA2_chr12_12496109: 5′-TGTATCTGAGGTGAACCACTGC-3′
FW_Control_gRNA2_chr19_18226209: 5′-GACACCCAGAGTTCATTGTGAG-3′
RV_Control_gRNA2_chr19_18226209: 5′-GCTTTTGGTGAATCAGAGCATCC-3′
FW_Control_gRNA2_chr14_96046026: 5′-CATGGTGTGGACTGTTTCTGTT-3′
RV_Control_gRNA2_chr14_96046026: 5′-AGGTTTTGCCAAGCTCTACACA-3′
FW_Control_gRNA2_chr5_2514714: 5′-CACAGGGTTTGGTGTTTGTAGG-3′
RV_Control_gRNA2_chr5_2514714: 5′-TGATCTGGCTTGACTTGCTTCC-3′
FW_Control_gRNA2_chr8_10811909: 5′-AGGAAGACAAATCCTAAGACAGCC-3′
RV_Control_gRNA2_chr8_10811909: 5′-AAGTTTTACGTTTTCCTCGACCAC-3′
FW_Control_gRNA2_chr3_122079794: 5′-AGATAAGCCAAGCCAGATTCTTCA-3′
RV_Control_gRNA2_chr3_122079794: 5′-CTGCTTGGAATTCCCACATTACTC-3′
FW_Control_gRNA2_chr7_2644811: 5′-GTTTATAGACAACCCACTGAGCAC-3′
RV_Control_gRNA2_chr7_2644811: 5′-CTCATTGGCACTACTGCAGAGAAA-3′
FW_Control_gRNA2_chr15_54082949: 5′-ACTCATCTTAAGGAACCGTCGC-3′
RV_Control_gRNA2_chr15_54082949: 5′-CATAGCCACATCCAAAGGTCCA-3′

**Karyotyping analysis**. Lineage depleted cord blood cells were thawed and CD34$^+$ CD38$^-$ cells were sorted using CD34 APC-Cy7 (clone 581, custom conjugation) and CD38 PE-Cy7 (335825, HB7). Cells were pre-cultured for 36–48 h in serum-free media and subsequently electroporated as described above. After overnight incubation, media was changed to IMDM (Thermo Fisher, 12440061) with 10% FBS (Sigma, 15A085), 1× L-Glutamine (Thermo Fisher, 25030081), 1× Penicillin–Streptomycin (Thermo Fisher, 15140122) and the following cytokines (all from Miltenyi Biotec): FLT3L (100 ng/mL), G-CSF (10 ng/mL), SCF (100 ng/mL), TPO (15 ng/mL), and IL-6 (10 ng/mL) to allow for cell expansion. Subsequently, karyotyping of chromosomes was performed according to standard procedures. Metaphase slides were prepared, then banded with trypsin and stained with Leishman's stain. The G-banded slides were scanned and metaphases captured using an automated imaging system (Meta-Systems). Metaphases were analyzed using Ikaros image analysis software (Meta-Systems). For each condition, 20 metaphases were analyzed by G-banded karyotyping for numerical and structural abnormalities.

**Statistical analysis**. Error bars represent standard deviations. Statistical significance was assessed using two-tailed unpaired student's *t* test.

**Life sciences reporting summary**. Additional information on experimental design is available in the Nature Research Reporting Summary linked to this article.

**Reporting summary**. Further information on research design is available in the Nature Research Reporting Summary linked to this article.

## Data availability
All datasets generated in this study are available within the paper or from the corresponding author upon reasonable request. Full length gel pictures and western assay can be found in Supplementary Fig. 12.

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

## Acknowledgements

We thank M. DSouza for support with mouse work; T. Velauthapillai, S. Ng, A. AuYeung, N. Simard and S. Zhao at the SickKids-UHN Flow and Mass Cytometry Facility for assistance with flow cytometry; B. Apresto at the The Centre for Applied Genomics (SickKids) for Sanger sequencing; A. Smith at the Cancer Cytogenetics Laboratory (UHN) for karyotyping; M. Peralta and C. Cimafranca at the Pathology Research Program (UHN) for assistance with histology; M. Bartolini at the Advanced Optical Microscopy Facility (AOMF) for slide scanning; the labs of S. Chan and F. Notta for shared equipment. We thank M. Anders, J. Kennedy and J. Wang for comments on the manuscript. E.W. is supported by a long-term fellowship from the Human Frontier Science Program and a Banting postdoctoral fellowship. L.D.S is supported by National Institute of Health CA034196, AI132963 and DK104218. Work in J.E.D.'s laboratory is supported by grants from the Canadian Institutes for Health Research, Ontario Institute for Cancer Research through funding provided by the Government of Ontario, Terry Fox Foundation Research Institute, Canadian Cancer Society and the University of Toronto's Medicine by Design initiative, which receives funding from the Canada First Research Excellence.

## Author contributions

E.W., J.E.D. and E.L. conceived the project, supervised research, and wrote the paper. E.W., O.I.G. and E.L. analyzed the experiments. E.W., M.A., S.S., L.S. and E.L. performed the in vitro and in vivo experiments. J.L.M. assisted with intrafemoral injections. G.K. assisted with western assay. J.A. assisted with morphology analysis. O.I.G. assisted with single cell in vitro assays and methylcellulose colony formation assays. L.D.S. provided NSGW41 mice.

## Competing interests

The authors declare no competing interests.
