## [Peer Review File · Nature Communications]

Reviewers' comments:

Reviewer #1 (Remarks to the Author):

Wagenblast et al in "Functional Profiling of Single CRISPR/Cas9-Edited Human Long- Term Hematopoietic Stem Cells" describe their work using genome editing of human HSCs and HSPCs using the ribonucleoprotein complex method of Cas9 based genome editing to study the biologic function of the different GATA1 isoforms (short and long). They perform the experiments by sorting human cells into LT-HSCs, ST-HSCs, MEP... by flow cytometry using state of the art markers. They then subject these enriched populations to editing by RNP via either NHEJ based deletion of exon 2 or HDR inactivation of a splice site. They then analyze the resulting populations using both in vitro and limiting dilution xenotransplant assays for LT-HSCs. The experiments are well described, well controlled, and well presented with one weakness being in certain engraftment experiments, the N is sometimes only 3 mice.

The authors make 3 major conclusions:

1. That human LT-HSCs can be edited and engrafted. This conclusion is supported by their data but is not novel. Multiple papers from the Naldini lab have performed similar sorting experiments to demonstrate this fact and papers from the Bauer and Porteus labs demonstrate functional engraftment of edited (both NHEJ and HR edited) LT-HSCs as determined by serial transplantation. These prior publications need to be better referenced and included in both the introduction and discussion. This work adds to that literature in a rigorous fashion but is not novel and the authors, while using state of the art genome editing, strategies do not provide any technical advances. The limiting dilution engraftment experiments for edited LT-HSCs have not been published before but did not reveal any novel results that were not previously found in prior studies.
2. The second point is to model DS-AMKL. The authors are appropriately balanced in describing how the editing approach followed by xeno-transplantation is a first step towards a better understanding of this disease. The authors should reference other leukemia disease modeling work using genome editing including that published by the Ebert, Mulligan and Cleary labs among others.
3. The third major point is to use editing to better understand the differential role of the GATA1 isoforms in differentiation from LT-HSCs. The authors provide rigorous and informative data on the lineage bias that occurs when specifically altering the expression of each isoform. This focused editing strategy is an important demonstration to how editing, when thoughtfully applied, can be used to understand stem cell biology, particularly hematopoietic stem cell biology.

The off-target analysis shown in Supp Figure 4 is interesting but of somewhat limited depth. The use of amplicon sequencing on individual clones on a reasonable set of potential off-target sites is a nice first pass. Deep amplicon sequencing of potential off-target sites from a population would be better but I don't believe would add significantly to the work. The clear linkage between the known biology of GATA1, the phenotypic results found, and the likelihood that the results are from off-target InDels is extremely low. In an era when resources are not unlimited, performing more extensive off-target analysis, while almost a reflexive thing that reviewers request, is not needed to support the conclusions of this paper.

A supplementary figure showing the junction sequences after deleting exon 2 would be of interest to better understanding NHEJ based generation of deletions occurs in HSCs and HSPCs.

Otherwise, other than perhaps increasing the N in certain experiments to further establish the quantitative effects of the interventions, I have no specific suggestions that would result in a significant improvement in the work.

Reviewer #2 (Remarks to the Author):

The manuscript by Wagenblast et al. focuses on evaluation of CRISPR/Cas9 editing of human umbilical cord blood-derived long-term hematopoietic stem cells (LT-HSC). Specifically, the authors use a slightly modified version of previously published approaches to directly edit LT-HSC, ST-HSC and MEP using CRISPR/Cas9 and functionally interrogate edited cells using both in vitro colony forming assays and long-term xenotransplantation assays into NSG and NSGW41 mice. Using GATA1 locus editing to induce expression of short or long isoform of GATA1, the authors document functional consequences of switching to GATA1-Short isoform only expression, which is associated with an increased number of megakaryocyte colonies and megakaryocyte expansion in long-term transplant assays. All experiments are well executed and the data are clearly presented. This report feels somewhat technical in nature without many new conceptual insights provided beyond the functional consequences of GATA1-Short isoform expression.

1. The authors nicely demonstrate an experimental way to toggle between GATA1-Short versus Long isoform expression in umbilical cord blood LT-HSC, ST-HSC and MEP populations. Have the authors attempted to use CRISPR/Cas9 editing to model the specific mutations in exon 2 of GATA1 observed in patients with Down Syndrome AMKL that lead to GATA1-Short isoform expression in umbilical cord blood LT-HSC, and examine the functional consequences on megakaryocyte proliferation? In addition, have the authors attempted to do this in adult LT-HSC to further evaluate their hypothesis of developmental stage-specific effects of certain genetic lesions such as GATA1 mutations?
2. I find the calculations of % single cell CRISPR efficiency in the manuscript somewhat vague. How many colonies were picked and sequenced in each experiment? Was next generation sequencing performed in the bulk population prior to/in parallel to single cell sorting?
3. Similarly, regarding calculations of % mice with engraftment and CRISPR editing (Figure 3a, Supplemental Figure 5b, c), how many mice were used in total for each cord blood donor? Tables with raw data per mouse should be provided.
4. Regarding off-target analysis (Supplemental Figure 4a), did the authors observe any karyotypic abnormalities in their edited LT-HSC, particularly in cases where multiple gRNAs were used? A very small number of off-target sites was examined (in the case of some of the gRNAs only 2-3 possible sites) and should be expanded.
5. Did the authors observe any changes in the immunophenotype of their sorted LT-HSC and ST-HSC with the 36-48hr culture prior to electroporation?

Reviewer #3 (Remarks to the Author):

In this manuscript, Wagenblast and colleagues investigate CRISPR/Cas9-mediated genomic editing efficiencies and their functional consequences in human hematopoietic stem and progenitor cells. The novelty and interest in this work is two-fold: the authors describe and functionally characterize genomic editing in purified LT-HSCs and they manipulate endogenous GATA1 expression and show the effects on cell fate.

In their approach, bulk umbilical cord-derived CD34+ cells are first sorted by cell surface markers into phenotypic LT-HSC, ST-HSC, and MEP populations, which are subsequently subjected to CRISPR/Cas9-based genome editing. In this way, they are able to detect editing frequencies in LT-HSCs and use these edited LT-HSCs in single cell in vitro differentiation studies and near-clonal xenotransplantation experiments. To carry out this study, the authors have chosen the transcription factor GATA1 as their

target gene. They show that forced expression of the short isoform of GATA1 by targeted deletion leads to increased production of megakaryocytes with variable effect on B-lymphocytes and erythroid cells.

This is a valuable study that should be published, but a few issues discussed below should be addressed through modifications of the manuscript or through the inclusion of additional data.

Major Comments:

- The authors cite a link between GATA1 mutations and Down syndrome associated AMKL and postulate that forced expression of GATA1s may be the cause of the increased megakaryopoiesis. The authors should discuss how the observed phenotypes are similar to or distinct from germline GATA1 mutations described in Diamond Blackfan anemia that result in almost exclusive GATA1s expression, but that have not been reported to increase megakaryocytes (PMID 22706301, 24453067, 24952648). It would be valuable to discuss how the presented data could fit with these clinical observations.

- The NSGW41 mouse model used in Figure 4 shows a clear increase in megakaryocytes and loss of erythroid cells upon forced GATA1s expression. The presented data from the NSG mouse in Figure 3 is more challenging to interpret. There is a significant increase in phenotypic B cells and in total cell number in the GATA1s mice. The authors state that the increase in B-cells comes "at the expense of myeloid cells", but the absolute number of myeloid cells in Figure 3e appears equivalent in the experimental and control animals. This should be clarified. The authors suggest that "lymphoid bias" may be an explanation for the doubling of B-cell number in the GATA1s NSG mice. This should be further explained. Altogether, the data presented in Figure 3 does not add much to the data in Figure 4 and may not be necessary to include in the main figures.

- In addition to the surface phenotypic assessment of erythroid and megakaryocytic markers in the NSGW41 mice shown in Figure 4, have the authors attempted to sort these populations and examine morphology as other investigators have done previously in these mouse models (PMID 27618723, 28568895)? If possible, it would be valuable to assess for alterations in the morphology of hematopoietic populations from both the erythroid and megakaryocytic lineages.

Minor Comments:

- Lane 1 of the Western blot in Figure 1c has a reported ratio of GATA1 short/long of 0.2 and lane 2 has a ratio of 0.11. However, there appears to be no GATA1s band in lane 2.

- In the main text, the authors write, "a 5-fold increase in megakaryocytic output was observed in the RF and BM of mice transplanted with GATA1-short edited LT-HSCs." However, in Figure 4c, this increase appears to be at most 3-fold.

- The authors sort for phenotypic LT-HSCs and then perform nucleofection 36-48 hours later with xenotransplantation or in vitro differentiation that begins the following day. It would be informative to see FACS plots of the LT-HSC population after 48-72 hours in culture to see what percentage of cells maintain their LT-HSC phenotypic markers.

Point-by-Point Response:

We would like to thank the reviewers for their constructively critical comments. We have addressed all the reviewer's concerns and updated the main text, figures and supplementary materials. We feel that these have helped us to strengthen the manuscript. Our detailed responses follow.

Reviewer 1:

Wagenblast et al in "Functional Profiling of Single CRISPR/Cas9-Edited Human Long-Term Hematopoietic Stem Cells" describe their work using genome editing of human HSCs and HSPCs using the ribonucleoprotein complex method of Cas9 based genome editing to study the biologic function of the different GATA1 isoforms (short and long). They perform the experiments by sorting human cells into LT-HSCs, ST-HSCs, MEP... by flow cytometry using state of the art markers. They then subject these enriched populations to editing by RNP via either NHEJ based deletion of exon 2 or HDR inactivation of a splice site. They then analyze the resulting populations using both in vitro and limiting dilution xenotransplant assays for LT-HSCs. The experiments are well described, well controlled, and well presented with one weakness being in certain engraftment experiments, the N is sometimes only 3 mice.

We apologize if the reviewer misinterpreted the presentation of the data. To be clear, the *in vivo* xenotransplantation assays were performed with 3 cohorts of mice. In total 30 NSG mice engrafted with human LT-HSC with validated GATA1-Short genotype and 23 mice transplanted with LT-HSC with validated Control gRNA genotype were included in the analysis of the xenotransplantation experiments in NSG mice. For the xenotransplantation experiments in NSGW41 mice, a total of 24 and 22 mice transplanted with LT-HSC with validated GATA1-Short genotype and Control gRNA genotype were included, respectively. To avoid any confusion, we included a new data table (Supplementary Fig. 7d, 9d) that displays the number of mice we utilized in each cohort and the number of mice that we included for analysis after validation of the CRISPR/Cas9 genotype.

The authors make 3 major conclusions:

1. That human LT-HSCs can be edited and engrafted. This conclusion is supported by their data but is not novel. Multiple papers from the Naldini lab have performed similar sorting experiments to demonstrate this fact and papers from the Bauer and Porteus labs demonstrate functional engraftment of edited (both NHEJ and HR edited) LT-HSCs as determined by serial transplantation. These prior publications need to be better referenced and included in both the introduction and discussion. This work adds to that literature in a rigorous fashion but is not novel and the authors, while using state of the art genome editing strategies do not provide any technical advances. The limiting dilution engraftment experiments for edited LT-HSCs have not been published before but did not reveal any novel results that were not previously found in prior studies.

To our knowledge, we are presenting the first manuscript that shows the feasibility of isolating LT-HSCs by sorting with the best available markers and directly using them for CRISPR/Cas9 editing by electroporation. Using our method, functional interrogation of LT-HSCs can be carried out within 16 days *in vitro* using single cell differentiation assays and 12 weeks *in vivo* using near clonal xenotransplantations. Papers from the Porteus, Naldini and Bauer labs elegantly utilize CD34⁺ HSPCs for CRISPR/Cas9 electroporation. These studies interrogate the functional effect of CRISPR/Cas9 edited LT-HSCs by carrying out up to 20 week-long xenotransplantation assays. We indeed cite all of these publications in the introduction section of the manuscript. We wonder if the reviewer may be referring to the Schirotti, Naldini et al. paper¹, specifically. If this would be the case, the contention that similar sorting experiments have been performed in this study is not accurate. The authors electroporated CD34⁺ HSPCs and 24 hours afterwards sorted CD34⁺CD133⁺CD90⁺ and CD34⁺CD133⁺CD90⁻ cells for single cell transcriptomics and methylcellulose colonies. Importantly, no *in vivo* functional assays appear to have been performed with these sorted cells. This experimental workflow is different compared to ours, since we prospectively isolate LT-HSCs (CD34⁺CD38⁻CD45RA⁻CD90⁺CD49f⁺) for direct CRISPR/Cas9 editing, providing a pool of relatively pure LT-HSC with a high frequency of editing. We believe that by combining state of the art sorting markers and single cell *in vitro* and *in vivo* assays our methodology has merit, because it provides a more direct and rapid assessment of LT-HSC function.

2. The second point is to model DS-AMKL. The authors are appropriately balanced in describing how the editing approach followed by xenotransplantation is a first step towards a better understanding of this disease. The authors should reference other leukemia disease modeling work using genome editing including that published by the Ebert, Mulligan and Cleary labs among others.

We apologize for omitting valuable references that place our work in the context of the rapidly evolving area of human hematopoietic disease modeling using CRISPR/Cas9 technology. To address this concern, we have included additional references describing other leukemia disease modeling work in the introduction section of the manuscript.

3. The third major point is to use editing to better understand the differential role of the GATA1 isoforms in differentiation from LT-HSCs. The authors provide rigorous and informative data on the lineage bias that occurs when specifically altering the expression of each isoform. This focused editing strategy is an important demonstration to how editing, when thoughtfully applied, can be used to understand stem cell biology, particularly hematopoietic stem cell biology.

We thank the reviewer for their kind comments. It was our goal to increase the efficiency of CRISPR/Cas9 editing in purified LT-HSCs and provide a blueprint for the rigorous directed study of these cells to model hematological diseases.

The off-target analysis shown in Supp Figure 4 is interesting but of somewhat limited depth. The use of amplicon sequencing on individual clones on a reasonable set of potential off-target sites is a nice first pass. Deep amplicon sequencing of potential off-

target sites from a population would be better but I don't believe would add significantly to the work. The clear linkage between the known biology of GATA1, the phenotypic results found, and the likelihood that the results are from off-target InDels is extremely low. In an era when resources are not unlimited, performing more extensive off-target analysis, while almost a reflexive thing that reviewers request, is not needed to support the conclusions of this paper.

While we agree with the reviewer's comment that next-generation whole genome sequencing would be a more global approach to assess off target effects, we extended our off-target analysis to include more potential off-target sites. On average, we now analyze between 9-12 off-target sites that range between 2-4 mismatches compared to the gRNA sequence. We have not detected any off-target cleavage in any of the CRISPR/Cas9-edited colonies. In addition, we included the following sentence in the manuscript: "Although no whole genome sequencing was performed, the likelihood that these results are due to off-target cleavage are extremely low."

A supplementary figure showing the junction sequences after deleting exon 2 would be of interest to better understanding NHEJ based generation of deletions occurs in HSCs and HSPCs.

These junction sequences can be seen in Supplementary Fig. 7g.

Otherwise, other than perhaps increasing the N in certain experiments to further establish the quantitative effects of the interventions, I have no specific suggestions that would result in a significant improvement in the work.

This comment was addressed above. Please refer to our first comment/answer.

Reviewer 2:

The manuscript by Wagenblast et al. focuses on evaluation of CRISPR/Cas9 editing of human umbilical cord blood-derived long-term hematopoietic stem cells (LT-HSC). Specifically, the authors use a slightly modified version of previously published approaches to directly edit LT-HSC, ST-HSC and MEP using CRISPR/Cas9 and functionally interrogate edited cells using both in vitro colony forming assays and long-term xenotransplantation assays into NSG and NSGW41 mice. Using GATA1 locus editing to induce expression of short or long isoform of GATA1, the authors document functional consequences of switching to GATA1-Short isoform only expression, which is associated with an increased number of megakaryocyte colonies and megakaryocyte expansion in long-term transplant assays. All experiments are well executed and the data are clearly presented. This report feels somewhat technical in nature without many new conceptual insights provided beyond the functional consequences of GATA1-Short isoform expression.

1. The authors nicely demonstrate an experimental way to toggle between GATA1-Short versus Long isoform expression in umbilical cord blood LT-HSC, ST-HSC and MEP populations. Have the authors attempted to use CRISPR/Cas9 editing to model the specific mutations in exon 2 of GATA1 observed in patients with Down Syndrome AMKL that lead to GATA1-Short isoform expression in umbilical cord blood LT-HSC, and examine the functional consequences on megakaryocyte proliferation? In addition, have the authors attempted to do this in adult LT-HSC to further evaluate their hypothesis of developmental stage-specific effects of certain genetic lesions such as GATA1 mutations?

All patient-associated mutations of GATA1 in Down Syndrome associated transient leukemia and AMKL lead to the sole expression of the GATA1-Short isoform. Thus, the CRISPR/Cas9 mediated excision of exon 2 directly mimics what is observed in patients in terms of isoform expression. However, in order to satisfy the reviewer's comment concerning specific mutations seen in these patients, we have now included genetically verified single-cell derived colonies from LT-HSCs, ST-HSCs and MEPs where only one gRNA targeted the 5' splice site of exon 2 (Supplementary Fig. 6a-c). This creates insertions/deletions at the 5' splice junction of exon 2, which resemble frequent mutations that are seen in patients with Down Syndrome associated transient leukemia and AMKL². We see an increase in the number of megakaryocyte colonies in GATA1 splice junction edited LT-HSCs (M, Meg) and ST-HSCs and MEPs (M, E, Meg), comparable to what we see when exon 2 is completely excised using CRISPR/Cas9. Whereas, we are interested in assessing the phenotype of GATA1-Short mutations during human ontology, we have not performed these experiments in bone marrow derived LT-HSCs to date.

2. I find the calculations of % single cell CRISPR efficiency in the manuscript somewhat vague. How many colonies were picked and sequenced in each experiment? Was next generation sequencing performed in the bulk population prior to/in parallel to single cell sorting?

In order to clarify the percentage of single cell CRISPR efficiency for the reader, we have now included a table in Supplementary Fig. 5a, which indicates the number of single cells sorted, the number of single cell derived colonies and the number of CRISPR/Cas9-edited colonies for each cell type. We hope these changes will clarify the amount of work we performed for these studies.

In parallel to the single cell sorting for *in vitro* differentiation, we performed Sanger sequencing of the remaining bulk cell population to validate correct CRISPR/Cas9 editing, in cases where enough cells remained. The editing efficiency in the bulk population as determined by TIDE was around 70-80% cleavage efficiency for Control and GATA1-Short edited cells and around 20% HDR efficiency for GATA1-Long. Percent of single cell CRISPR efficiency for Control and GATA1-Short edited colonies depicted in Fig. 2a show lower values, as only colonies that produced a shift in the PCR product as seen in Supplementary Fig. 4a, b (removal of the whole exon) were included.

3. Similarly, regarding calculations of % mice with engraftment and CRISPR editing (Figure 3a, Supplemental Figure 5b, c), how many mice were used in total for each cord blood donor? Tables with raw data per mouse should be provided.

We agree with the Reviewer 2 that our manuscript was unclear as to mouse numbers. In order to address these concerns, we updated our manuscript to include tables in Supplementary Fig. 7d and 9d, which demonstrate the number of mice injected, number of mice engrafted and number of engrafted mice with the desired CRISPR/Cas9 edit.

4. Regarding off-target analysis (Supplemental Figure 4a), did the authors observe any karyotypic abnormalities in their edited LT-HSC, particularly in cases where multiple gRNAs were used? A very small number of off-target sites was examined (in the case of some of the gRNAs only 2-3 possible sites) and should be expanded.

In response to this query, we undertook a new karyotyping analysis performed in CD34⁺CD38⁻ CRISPR/Cas9 electroporated cells. No structural abnormalities were seen in any of the conditions (Supplementary Fig. 5c). In addition, in our response to Reviewer 1 (see above), we have further addressed concerns regarding the specificity of targeting by extended our off-target analysis to include more potential off-target sites. On average, we now analyze between 9-12 off-target sites that range between 2-4 mismatches compared to the gRNA sequence. We have not detected any off-target cleavage in any of the CRISPR/Cas9-edited colonies. In addition, we included the following sentence in the manuscript: “Although no whole genome sequencing was performed, the likelihood that these results are due to off-target cleavage are extremely low.” We believe this additional data reaches an acceptable threshold to address the reviews’ concerns regarding off-target effects.

5. Did the authors observe any changes in the immunophenotype of their sorted LT-HSC and ST-HSC with the 36-48hr culture prior to electroporation?

It is known that the immunophenotype of primary human lineage depleted CB cells is not stable after long-term *in vitro* culture. To address whether the LT-HSC

immunophenotype is altered during our short-term culture conditions, we sorted LT-HSCs, ST-HSCs, CMPs and MEPs and cultured them *in vitro* for 24, 48 and 72 hours in order to evaluate the expression of the cell surface markers. Overall, the phenotypic profile of LT-HSCs largely remained the same (CD34⁺CD38⁻CD45RA⁻CD90⁺CD49f⁺). The phenotypic profiles of ST-HSCs, CMPs and MEPs were a bit less stable. In particular, MEPs gained FLT3 expression within 24 hours of culture, and this may be due to the presence of FLT3 ligand in the media. Please see representative flow cytometry plots in new Supplementary Fig. 3a, b.

Reviewer 3:

In this manuscript, Wagenblast and colleagues investigate CRISPR/Cas9-mediated genomic editing efficiencies and their functional consequences in human hematopoietic stem and progenitor cells. The novelty and interest in this work is two-fold: the authors describe and functionally characterize genomic editing in purified LT-HSCs and they manipulate endogenous GATA1 expression and show the effects on cell fate.

In their approach, bulk umbilical cord-derived CD34+ cells are first sorted by cell surface markers into phenotypic LT-HSC, ST-HSC, and MEP populations, which are subsequently subjected to CRISPR/Cas9-based genome editing. In this way, they are able to detect editing frequencies in LT-HSCs and use these edited LT-HSCs in single cell in vitro differentiation studies and near-clonal xenotransplantation experiments. To carry out this study, the authors have chosen the transcription factor GATA1 as their target gene. They show that forced expression of the short isoform of GATA1 by targeted deletion leads to increased production of megakaryocytes with variable effect on B-lymphocytes and erythroid cells.

This is a valuable study that should be published, but a few issues discussed below should be addressed through modifications of the manuscript or through the inclusion of additional data.

Major Comments:

The authors cite a link between GATA1 mutations and Down syndrome associated AMKL and postulate that forced expression of GATA1s may be the cause of the increased megakaryopoiesis. The authors should discuss how the observed phenotypes are similar to or distinct from germline GATA1 mutations described in Diamond Blackfan anemia that result in almost exclusive GATA1s expression, but that have not been reported to increase megakaryocytes (PMID 22706301, 24453067, 24952648). It would be valuable to discuss how the presented data could fit with these clinical observations.

Diamond Blackfan anemia is a hereditary genetic disease characterized by low red blood cell counts and usually presents in infancy. Germline mutations of GATA1 have recently been described in patients with Diamond Black fan anemia³⁻⁵. These patients have almost exclusive GATA1-Short expression and no increase in megakaryocytes have been reported in the abovementioned studies. We hypothesize that in these patients due to the germline nature of GATA1 mutations, in the course of fetal blood development some unidentified compensatory mechanism occurred to allow for more normalized megakaryocytic development. It is interesting to note that these patients often have increased miscarriages⁶. We now have included a paragraph about this within the discussion section of the manuscript.

The NSGW41 mouse model used in Figure 4 shows a clear increase in megakaryocytes and loss of erythroid cells upon forced GATA1s expression. The presented data from the NSG mouse in Figure 3 is more challenging to interpret. There is a significant increase

in phenotypic B cells and in total cell number in the GATA1s mice. The authors state that the increase in B-cells comes “at the expense of myeloid cells”, but the absolute number of myeloid cells in Figure 3e appears equivalent in the experimental and control animals. This should be clarified. The authors suggest that “lymphoid bias” may be an explanation for the doubling of B-cell number in the GATA1s NGS mice. This should be further explained. Altogether, the data presented in Figure 3 does not add much to the data in Figure 4 and may not be necessary to include in the main figures.

It is known that the NSG xenotransplantation model has certain limitations regarding human hematopoietic lineage output. Indeed, the model is biased toward human lymphoid lineage development.

In our experiments in NSG mice the percentage of CD19⁺ B-lymphoid cells significantly increases in GATA1-Short edited LT-HSCs injected mice compared to control edited LT-HSCs injected mice. This increase in phenotypic B-cells is also seen in term of absolute numbers of CD19⁺ B-lymphoid between GATA1-Short edited LT-HSCs injected mice compared to control. On the other hand, there is a slight, but statistically significant decrease in the percentage, but not in the absolute numbers of CD33⁺ Myeloid cells in GATA1-Short edited LT-HSCs injected mice compared to control (from 10% to 8%). In general, in NSG mice myeloid grafts are small and thus a difference in absolute CD33⁺ cell numbers might not easily be seen in the analysis. It is true that the increase in B-cells outweighs the rather small effect in the percentage of CD33⁺ Myeloid cells and thus we removed the comment: “at the expense of myeloid cells”.

Finally, because NSG mice are more frequently used than NSGW41 mice in human xenotransplantation experiments, we believe it is beneficial to present the NSG xenotransplantation data.

In addition to the surface phenotypic assessment of erythroid and megakaryocytic markers in the NSGW41 mice shown in Figure 4, have the authors attempted to sort these populations and examine morphology as other investigators have done previously in these mouse models (PMID 27618723, 28568895)? If possible, it would be valuable to assess for alterations in the morphology of hematopoietic populations from both the erythroid and megakaryocytic lineages.

We utilized the right femur (RF) of genetically verified control and GATA1-Short edited LT-HSCs injected NSGW41 mice and flow cytometry sorted GlyA⁺CD45⁻ erythroid cells and CD41⁺ megakaryocytic lineage derived cells. Sufficient numbers of CD41⁺CD45⁻ cells could not be detected after the freeze/thaw cycle of stored RF or BM and thus only bulk CD41⁺ cells were sorted, regardless of CD45 staining. GlyA⁺CD45⁻ cells were sorted from control and GATA1-Short edited LT-HSCs injected NSGW41 mice from the RF. Cells were cytopinned and stained with Giemsa. More immature forms of erythroid cells and fewer enucleated erythrocytes are visible in GATA1-Short edited LT-HSCs injected NSGW41 mice compared to control (Supplementary Fig. 11 a,b).

CD41⁺ cells were sorted from control and GATA1-Short edited LT-HSCs injected NSGW41 mice from the BM. Cells with variable amounts of blue-greyish cytoplasm, sometimes with a few vacuoles, and with fine granules and pleomorphic nuclei were frequently seen (Supplementary Fig. 11 c). Cells with cytoplasmic pseudopod

formation, sometimes exhibiting distinct nucleoli were also observed with less frequency. Finally, few large binucleated cells were observed. No striking differences in morphology were observed between CD41⁺ cells from GATA1-Short edited LT-HSCs injected NSGW41 mice compared to control.

Minor Comments:

- Lane 1 of the Western blot in Figure 1c has a reported ratio of GATA1 short/long of 0.2 and lane 2 has a ratio of 0.11. However, there appears to be no GATA1s band in lane 2.

The Western assay in Figure 2c was performed using the ProteinSimple Wes, a digital capillary-based platform with pg-level sensitivity. The ratios are calculated based on the intensity of the signals for GATA1-Short/GATA1-Long. We have confirmed that the reported ratios are correct as presented within the manuscript. We think the confusion concerning this figure derives from the lack of an observable band in lane 2. This is in part due to the exposure time that was chosen for the manuscript figure. There is indeed a faint band in lane 2, however, protein amount and thus intensity of the bands in the control sample is higher than in the GATA1-Long sample. Because of lower protein amounts in GATA1-Long, the GATA1-Short band within the GATA1-Long sample is quite dim and difficult to see. The advantage of using the Wes system is that the assays are quantitative over multiple exposure times and the results can easily be calculated with the included software.

In the main text, the authors write, “a 5-fold increase in megakaryocytic output was observed in the RF and BM of mice transplanted with GATA1-short edited LT-HSCs.” However, in Figure 4c, this increase appears to be at most 3-fold.

We thank the reviewer for catching our mistake. The correct fold-change increase of the percentage of CD41 megakaryocytic cells in Control vs. GATA1-Short is 3.2 for RF and 3.7 for BM. We have amended the text accordingly to 3-fold.

The authors sort for phenotypic LT-HSCs and then perform nucleofection 36-48 hours later with xenotransplantation or in vitro differentiation that begins the following day. It would be informative to see FACS plots of the LT-HSC population after 48-72 hours in culture to see what percentage of cells maintain their LT-HSC phenotypic markers.

In addition, in our response to Reviewer 2 (see above), it is known that the immunophenotype of primary human lineage depleted CB cells is not stable after *in vitro* culture. To address whether the LT-HSC immunophenotype is altered during our short-term culture conditions, we sorted LT-HSCs, ST-HSCs, CMPs and MEPs and cultured them *in vitro* for 24, 48 and 72 hours in order to evaluate the expression of the cell surface markers. Overall, the phenotypic profile of LT-HSCs largely remained the same (CD34⁺CD38⁻CD45RA⁻CD90⁺CD49f⁺). The phenotypic profiles of ST-HSCs, CMPs and MEPs were a bit less stable. In particular, MEPs gained FLT3 expression within 24 hours of culture, and this may be due to the presence of FLT3 ligand in the media. Please see representative flow cytometry plots in new Supplementary Fig. 3a, b.

References:

1. Schirotti, G. *et al.* Precise Gene Editing Preserves Hematopoietic Stem Cell Function following Transient p53-Mediated DNA Damage Response. *Cell Stem Cell* (2019). doi:10.1016/j.stem.2019.02.019
2. Rainis, L. *et al.* Mutations in exon 2 of GATA1 are early events in megakaryocytic malignancies associated with trisomy 21. *Blood* **102**, 981–986 (2003).
3. Ludwig, L. S. *et al.* Altered translation of GATA1 in Diamond-Blackfan anemia. *Nat. Med.* **20**, 748–753 (2014).
4. Parrella, S. *et al.* Loss of GATA-1 full length as a cause of Diamond-Blackfan anemia phenotype. *Pediatr Blood Cancer* **61**, 1319–1321 (2014).
5. Sankaran, V. G. *et al.* Exome sequencing identifies GATA1 mutations resulting in Diamond-Blackfan anemia. *J. Clin. Invest.* **122**, 2439–2443 (2012).
6. Giri, N., Reed, H. D., Stratton, P., Savage, S. A. & Alter, B. P. Pregnancy outcomes in mothers of offspring with inherited bone marrow failure syndromes. *Pediatr Blood Cancer* **65**, e26757 (2018).

REVIEWERS' COMMENTS:

Reviewer #1 (Remarks to the Author):

The authors have addressed my prior concerns. This work is a significant contribution to the literature on several fronts.

Reviewer #2 (Remarks to the Author):

The authors have addressed all of my concerns. The revised manuscript has been significantly strengthened.

Reviewer #3 (Remarks to the Author):

Wagenblast and colleagues have done a fantastic job of addressing all my concerns. Their revisions have helped add context to their work. This is an important paper for those interested in modeling and treating hematologic disorders using genome editing.